# Evaluating an accelerated forcing approach for improving computational efficiency in coupled ice sheet-ocean modelling

Qin Zhou[1], Chen Zhao[2], Rupert Gladstone[3], Tore Hattermann[4,5], David Gwyther[6], and Benjamin Galton-Fenzi[2,7,8]

[1]Akvaplan-niva AS, Tromsø, Norway
[2]Australian Antarctic Program Partnership, Institute for Marine and Antarctic Studies, University of Tasmania, Hobart, Australia
[3]Arctic Centre, University of Lapland, Rovaniemi, Finland
[4]Norwegian Polar Institute, Tromsø, Norway
[5]Energy and Climate Group, Department of Physics and Technology, The Arctic University - University of Tromsø, Norway
[6]Coastal and Regional Oceanography Lab, School of Biological, Earth and Environmental Sciences, UNSW Sydney, Sydney, NSW, Australia
[7]Australian Antarctic Division, Kingston, Australia
[8]The Australian Centre for Excellence in Antarctic Science, University of Tasmania, Hobart, Australia

**Correspondence:** Qin Zhou (qin@akvaplan.niva.no)

**Abstract.**

Coupled ice sheet-ocean models are increasingly being developed and applied to important questions pertaining to processes at the Greenland and Antarctic Ice Sheet margins, which play a pivotal role in ice sheet stability and sea level rise projections. One of the challenges of such coupled modelling activities is the timescale discrepancy between ice and ocean dynamics. This discrepancy, combined with the high computational cost of ocean models due to their finer temporal resolution, limits the time frame that can be modelled. In this study, we introduce an "accelerated forcing" approach to address the timescale discrepancy and thus improve computational efficiency in a framework designed to couple evolving ice geometry to ice shelf cavity circulation. This approach is based on the assumption that the ocean adjusts faster to imposed changes than the ice sheet, so the ocean can be viewed as being in a quasi-steady state that varies slowly over timescales of ice geometry change. By assuming that the mean basal melt rate during one coupling interval can be reflected by a quasi-steady state melt rate during a shortened coupling interval (equal to the regular coupling interval divided by a constant factor), we can reduce the ocean model simulation duration. We first demonstrate that the mean cavity residence time, derived from stand-alone ocean simulations, can guide the selection of suitable scenarios for this approach. We then evaluate the accelerated forcing approach by comparing basal melting response under the accelerated forcing with that under the regular forcing (without the accelerated forcing) based on idealized coupled ice sheet-ocean model experiments. Our results suggest that: the accelerated approach can yield comparable melting responses to those under the regular forcing when the model is subjected to steady far-field ocean conditions or time-varying conditions with timescales much shorter than the cavity residence time. However, it may not be suitable when the timescale of the accelerated ocean conditions is not significantly different from the cavity residence time. We have also discussed the limitations of applying the accelerated forcing approach to real-world scenarios, as it may not be applicable in coupled modelling studies addressing climate variability on sub-decadal, decadal, and mixed timescales, or in

fully coupled climate models with interactive ice sheets. Nevertheless, when appropriately applied, the accelerated approach can be a useful tool in process-oriented coupled ice sheet-ocean modelling or for downscaling climate simulations with a coupled ice sheet-ocean model.

## 1  Introduction

The Antarctic Ice Sheet represents the largest source of uncertainty in projections of sea level rise, with its contribution estimated to vary from -5 to 43 cm of sea level equivalent by 2100 under high emission scenarios (Seroussi et al., 2020, 2023). This uncertainty partly stems from the absence of ice sheet-ocean interactions in current sea level rise projections, which are based on stand-alone ice sheet models (Edwards et al., 2021; Seroussi et al., 2020). The interplay between the ice sheet and the ocean around Antarctica is a tightly coupled process and cannot be overlooked. Ocean-driven basal melting of floating ice

shelves, influenced by ocean currents and ice draft geometry, can trigger a non-linear response impacting ice-shelf buttressing, grounded ice velocity, grounding line movements, and ice sheet instabilities (Gladstone et al., 2012; Favier et al., 2014). Conversely, glacial meltwater from the ice shelves affects water mass transformation, sea ice formation and melting, alongside regional and global ocean circulation (Foldvik et al., 2004; Jourdain et al., 2017; Li et al., 2023), while subglacial drainage injection into ice shelf cavities drives strong local melt increases (Nakayama et al., 2021; Gwyther et al., 2023) and impacts

sea ice formation (Goldberg et al., 2023). Moreover, stand-alone ice sheet models lack physically sound methods to compute basal melt rates under newly ungrounded ice (Jourdain et al., 2020). Therefore, coupled ice sheet-ocean models are essential for capturing the complexity of ice sheet-ocean interactions and thus improve sea level rise projections.

Driven by these needs, recent years have witnessed significant developments in coupled ice sheet-ocean modelling. Some studies follow the guidelines of the 1st Marine Ice Sheet-Ocean Model Intercomparison Project (MISOMIP1) on idealized

domains (Asay-Davis et al., 2016; Favier et al., 2019; Zhao et al., 2022), while others are based on realistic, regionally-scaled domains like the Totten Glacier Area (Pelle et al., 2021; Van Achter et al., 2023; McCormack et al., 2021), the Thwaites Glacier (Seroussi et al., 2017), the Filchner-Ronne Ice Shelf (Timmermann and Goeller, 2017; Naughten et al., 2021). More recently, coupled ice sheet-ocean model configurations on the circumpolar scale or beyond, with cavities explicitly resolved, have begun to emerge (Smith et al., 2021; Pelletier et al., 2022; Siahaan et al., 2022). However, applying the circumpolar coupled ice-

ocean models to long-term simulations is heavily constrained by the timescale discrepancy between ice and ocean dynamics. The ice sheet timescale ranges from decades to millennia, while the ocean timescale spans from hours to decades. As a result, the typical timestep sizes are smaller for ocean models (seconds to minutes) compared to those for ice sheet models (days to months), making the ocean model more computationally demanding to run. These limitations prevent the coupled models from running a longer-term and larger ensemble of simulations, both of which are important for sea level rise projections.

This challenge of timescale discrepancies is not unique to coupled ice sheet-ocean modelling. A number of different climate-related disciplines utilising coupled modelling have encountered these issues of optimising the performance of a model system where individual components have varying response timescales, including atmosphere-ocean modelling (Sausen and Voss, 1996; Voss et al., 1998) and paleo-climate modelling incorporating ice sheets (Roberts et al., 2014; Lofverstrom et al., 2020).

Approaches have included "periodic synchronous coupling", where the outputs of the faster component are averaged over a short period of synchronous coupling and are then used to force the slower component(s) over a longer uncoupled period, and "asynchronous coupling", where the faster model is run for a shorter period during each coupling interval. In this context, "synchronous coupling" simply means that the elapsed modelled time, measured at the time of any exchange of coupled variables, is the same for each component. There is a broader definition that has been recently used in the ice sheet - ocean community (Goldberg et al., 2018; Gladstone et al., 2021), where "synchronous coupling" has been taken to mean that both fast and slow components update the coupling variables every fast timestep. Coupling synchronicity is especially important in the regional marine ice sheet - ocean modelling community where ice shelf cavity circulation is fully resolved by the ocean model and where the coupling region itself (the underside of the ice shelf) evolves with time.

In this study, we extend the concept of asynchronous coupling by introducing an approach of "time compression" or "accelerated forcing" to address the challenge of timescale discrepancy between the ocean and ice-sheet models. With this approach, the temporal scale of the ocean model is adjusted to be $\alpha$ times faster than the real-time temporal scale. $\alpha$ is referred to as the acceleration factor throughout the text. This approach shares the approach of a morphological acceleration factor used by the sediment transport modelling community, which effectively extends the morphological simulation duration by multiplying the changes in bed sediments by a constant factor (Lesse et al., 2004; Li et al., 2018; Morgan et al., 2020).

In the context of coupled ice sheet-ocean modelling, the accelerated forcing approach is based on the assumption that the ocean adjusts faster to imposed changes than the ice sheet, with the ocean viewed as being in a slowly varying quasi-steady state over the timescales that matter for ice sheet geometry. Note that the quasi-steady state here refers to the spun-up phase where the ocean model maintains a consistent average response to external forcings. Under this assumption, within the total ice draft change $\Delta z_d$, which includes contributions from ocean-driven change and ice-dynamics-driven change $\Delta z_{di}$, the ocean-driven draft change can be expressed as an integral of basal melt rate $M$ over the coupling time interval $T$, as

$$\Delta z_d = \int^T M dt + \Delta z_{di}. \tag{1}$$

The ocean-driven change can be further expressed as the time integral of a quasi-steady-state mean melt rate $\overline{M}^T$ over the coupling interval T, as

$$\int^T M dt = \overline{M}^T \cdot T. \tag{2}$$

By assuming that the mean melt rate $\overline{M}^T$ during the coupling interval $T$ can be approximated by a quasi-steady-state melt rate $\overline{M}^{T/\alpha}$ during a shortened coupling interval of $T/\alpha$, the ocean model simulation duration can be reduced from $T$ to $T/\alpha$, hereby accelerating the timescale of the ocean model by a factor of $\alpha$. Note that the superscripts $T$ and $T/\alpha$ denote the coupling intervals, not the exponents or powers of a number. In addition, to maintain the model's integrity under the accelerated approach, the timescales of the ocean model's boundary conditions should be also accelerated accordingly to accommodate the timescale change from $T$ to $T/\alpha$.

It is important to note that the above assumptions may not always hold true. However, our hypothesis proposes that this accelerated forcing approach remains valid under specific conditions - particularly when the quasi-steady state basal melting response is not sensitive to the timescale of varying boundary conditions that must be accordingly accelerated. This understanding provides a foundation upon which suitable scenarios for the approach can be determined. Specifically, it emphasizes the need to investigate how the basal melting in the ocean model responds to boundary conditions with varying timescales. In a regional coupled ice sheet-ocean model system, the ocean model is subject to a range of boundary conditions: changes in ice draft, heat and meltwater fluxes at the ice sheet-ocean interface, momentum, freshwater, and radiation fluxes at the atmosphere-ocean interface, and far-field ocean conditions. In this study, we only focus on the far-field ocean conditions and the ice draft change at the ice sheet-ocean interface, as these two factors predominantly control the cavity circulation and, thus, the basal melting response.

The far-field ocean conditions influencing ice sheet-ocean interactions around Antarctica range from seasonal, sub-decadal and decadal fluctuations (Dutrieux et al., 2014; Jenkins, 2016; Paolo et al., 2018; Jenkins et al., 2018), to longer, century-scale shifts associated with climate warming (Hellmer et al., 2012; Naughten et al., 2021). Given such vast variability, the systematic testing of the feasibility of the accelerated forcing approach becomes inefficient. Nevertheless, Holland (2017) suggests that the melting to time-varying ocean forcing is dictated by the relative magnitude of two timescales, the forcing timescale and the mean cavity residence time that is the characteristic time taken for the barotropic circulation to flush the entire cavity. The basal melting remains relatively stable when the cavity is subject to ocean conditions varying more rapidly than the cavity residence time, suggesting a scenario where the accelerated forcing approach might be applicable. However, the approach's applicability under ocean conditions, which vary slower than the cavity residence time, requires further experimental investigation. Following Holland (2017)'s study of exploring the melting response to time-varying ocean forcing, we will first use stand-alone ocean models with fixed ice cavities to identify the suitable scenarios for pragmatically applying the accelerated forcing approach.

The study is organized as follows: Section 2 briefly introduces the implementation of the accelerated forcing approach in the coupled ice sheet-ocean model system. Section 3 explores the basal melting to time-varying far-field ocean conditions to determine suitable scenarios for the approach with stand-alone ocean ~~experiments~~model experiments.. Section 4 assesses the approach across three scenarios with varied far-field ocean conditions using idealized coupled model setups. Lastly, Section 5 summarizes the findings and discusses the applicability and limitations of the approach.

## 2 Methodology and model description

### 2.1 The coupler: FISOC

The current study implements the accelerated forcing approach within the Framework For Ice Sheet–Ocean Coupling (FISOC). This flexible coupling framework, adopting the hierarchical modular structure of the Earth System Modelling Framework, allows the exchange of data between ice sheet and ocean models at the underside of the ice shelves (Gladstone et al., 2021). Figure 1 illustrates the workflow of the coupled ice sheet-ocean model system, both with and without the accelerated forcing approach. In the absence of the accelerated forcing approach, referred to as "regular forcing" within FISOC, the basal melt rates,

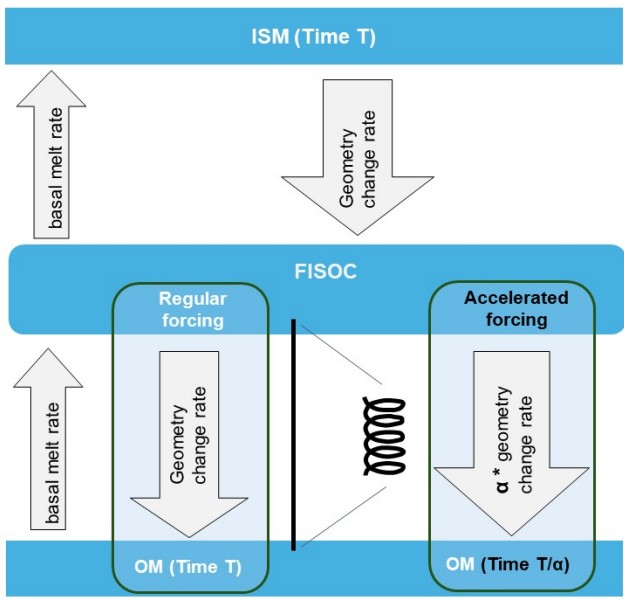

**Figure 1.** Data flow for the coupled ice sheet-ocean model system using FISOC, illustrating the differences between the regular forcing and the accelerated forcing approaches. ISM and OM stand for ice sheet model and ocean model, respectively. With the regular forcing approach, geometry change rate and basal melt rate are exchanged at regular time intervals. With the accelerated forcing approach, the geometry change rate passed to the ocean model is adjusted by multiplying it by the acceleration factor ($\alpha$).

calculated by the ocean model, are passed from the ocean model to the ice model, while geometry change rates, determined by the ice model, are passed from the ice model to the ocean model, as

$$\frac{dz_d}{dt}_{[O]} = \frac{dz_d}{dt}_{[I]}. \tag{3}$$

Here $z_d$ is the ice draft, and subscripts in square brackets indicate the representation of the same property within either the ice [I] or ocean [O] component. This exchange occurs at a coupling interval of $T$. Conversely, under the "accelerated forcing" approach with an acceleration factor $\alpha$, the boundary conditions imposed by the ice sheet model on the ocean model must be adjusted accordingly. Specifically, the geometry change rates received by the ocean model are amplified by a factor of $\alpha$, as

$$\frac{dz_d}{d(t/\alpha)}_{[O]} = \alpha \frac{dz_d}{dt}_{[O]} = \alpha \frac{dz_d}{dt}_{[I]}. \tag{4}$$

As the ocean model is run for a period of $T/\alpha$ for each coupling interval instead of $T$, the total change in ice geometry experienced by the ocean model during one coupling interval is the same under the accelerated forcing as the regular forcing. But the computational efficiency has been increased $\alpha$ times. It is important to note that throughout the text, we distinguish between *model time*, which refers to the ocean model's actual simulation time ($T/\alpha$ for one coupling interval), and *represented time*, which signifies the real-world time represented by the model, calculated as the model time multiplied by the acceleration factor ($T$ for one coupling interval).

## 2.2 The ice sheet model, Elmer/Ice

We use Elmer/Ice, a finite-element, dynamic ice sheet model (Gagliardini et al., 2013), as the ice model component in the coupled model system. The ice sheet model setup in this study is following Zhao et al. (2022). We use the Shallow Shelf Approximation (SSA*) solution, a variant of the L1L2 solution of Schoof (2010), to solve the shallow shelf approximation of the Stokes equations, which accounts for longitudinal and lateral stresses with an assumption of a simplified vertical shearing in the effective strain rate to represent fast-flowing ice streams and ice shelves. We apply a surface mass balance rate of 0.3 $\text{myr}^{-1}$ and assume a constant ice temperature in the ice model and zero heat flux across the ice-ocean boundary. The ice front location does not vary with time, and the ice mass loss due to calving disappears immediately without any freshwater flux into the ocean. We also apply a non-linear Weertman-type sliding relationship (Eq. (21) in Gagliardini et al. (2013)) for grounded ice, with a sliding parameter equal to 0.01 $\text{MPam}^{\frac{1}{3}}\text{a}^{\frac{1}{3}}$ and an exponent equal to $\frac{1}{3}$. The shelf regions are free slip.

## 2.3 The ocean models, FVCOM and ROMS

To increase the generality of our evaluation of the accelerated forcing approach, we conduct our main experiments using two different regional ocean models. The primary model is the Finite Volume Community Ocean Model (FVCOM) (Chen et al., 2003). While FVCOM is noted for its unstructured grid allowing geometric flexibility to resolve small-scale ice sheet-ocean interaction processes (Zhou and Hattermann, 2020), it is chosen here due to the authors' expertise with the model and its potential applications in future work. We also conduct selected experiments with the Regional Ocean Model System (ROMS) (Shchepetkin and McWilliams, 2005), which employs a structured Arakawa C-grid and has been widely used for resolving ice shelf cavities around Antarctica (Dinniman et al., 2007; Naughten et al., 2018; Galton-Fenzi et al., 2012; Richter et al., 2022). In addition to different grid structures, the two models differ in many aspects including numerical discretization, advection schemes, and mixing scheme parameterizations. Table 1 outlines some key differences in model characteristics between FVCOM and ROMS. However, it is worth noting that both models employ terrain-following vertical coordinates and share a number of similarities in resolving ice shelf cavities, including:

– Ice shelf-ocean thermodynamics are parameterized by the three-equation formulation following Jenkins et al. (2010). Specifically, in both models, values of $Cd = 0.0025$, $\Gamma_T = 0.05$, and $\Gamma_S = 0.0014$ are used for the drag coefficient and the turbulent heat and salt exchange coefficients, respectively.

– Both ocean models account for the thermodynamic effect of basal melting by imposing virtual heat and salt fluxes within a fixed geometry at each ocean model time step, to mimic the effects of basal melting, rather than employing an explicit volume flux at the ice-ocean interface.

– Ice shelf mechanical pressure is given by the density at the first layer of the model minus an assumed linear dependence of the density with depth, following Dinniman et al. (2007).

**Table 1.** Characteristics of the FVCOM and ROMS configurations used in this study.

|  | FVCOM | ROMS |
| --- | --- | --- |
| Horizontal grid | unstructured triangle grid | structured C-grid |
| Horizontal discretization | finite volumes | finite differences |
| Horizontal mixing scheme | Eddy closure parameterization | Laplacian mixing scheme |
| Vertical mixing scheme | Mellor and Yamada level 2.5 | K-Profile Parameterization |

– In coupled model setups, the grounding line movement is realized by the wet and dry scheme, allowing a passive water column under the grounded ice and an active water column under floating ice or in the open ocean. Note that the passive layer is very thin when dry and gets expanded when wet.

## 3   Stand-alone ocean model experiments

### 3.1   Experiment design

To increase the generality of our investigations of the melting response to time-varying far-field ocean conditions, we employ two model domains with different ice cavity geometries fixed in time for our stand-alone ocean model experiments. The first, as illustrated in Figure 2, is the Ice Shelf – Ocean Model Intercomparison Project (ISOMIP+) domain (Asay-Davis et al., 2016). It features a rectangular box bounded by $320\,\text{km} \leq \text{x} \leq 800\,\text{km}$ in the x direction and $0 \leq \text{y} \leq 80\,\text{km}$ in the y direction, with the grounding line position at $\text{x} = 460\,\text{km}$. The second domain features the same rectangular box but with a simplistic, wedge-shaped ice shelf in a flat-bottom ocean (Figure 2a). The wedge-shaped domain is implemented only with FVCOM, resulting in three model configurations: FVCOM-ISOMIP+, FVCOM-Wedge, and ROMS-ISOMIP+.

All configurations have a horizontal resolution of 2 km. The only external forcing in the model is a restoring forcing of far-field ocean conditions within 10 km of the lateral boundary of the domain ($790\,\text{km} \leq \text{x} \leq 800\,\text{km}$), as indicated by the green area in Figure 2a. The initial ocean properties and far-field ocean conditions consist of horizontally homogeneous temperature and salinity profiles that vary linearly with water depth:

$$T = T_0 + (T_b - T_0)\frac{z}{D} \tag{5}$$

and

$$S = S_0 + (S_b - S_0)\frac{z}{D}, \tag{6}$$

where $D = 720\,\text{m}$ is the maximum water depth, and $T_0$, $S_0$ and $T_b$, $S_b$ denote the surface and bottom values for temperature and salinity, respectively. Depending on the experiment, the temperature and salinity profiles used are either constant or oscillating over time. For constant profiles, as detailed in Table 2, we adopt the COLD and WARM profiles from Asay-Davis et al.

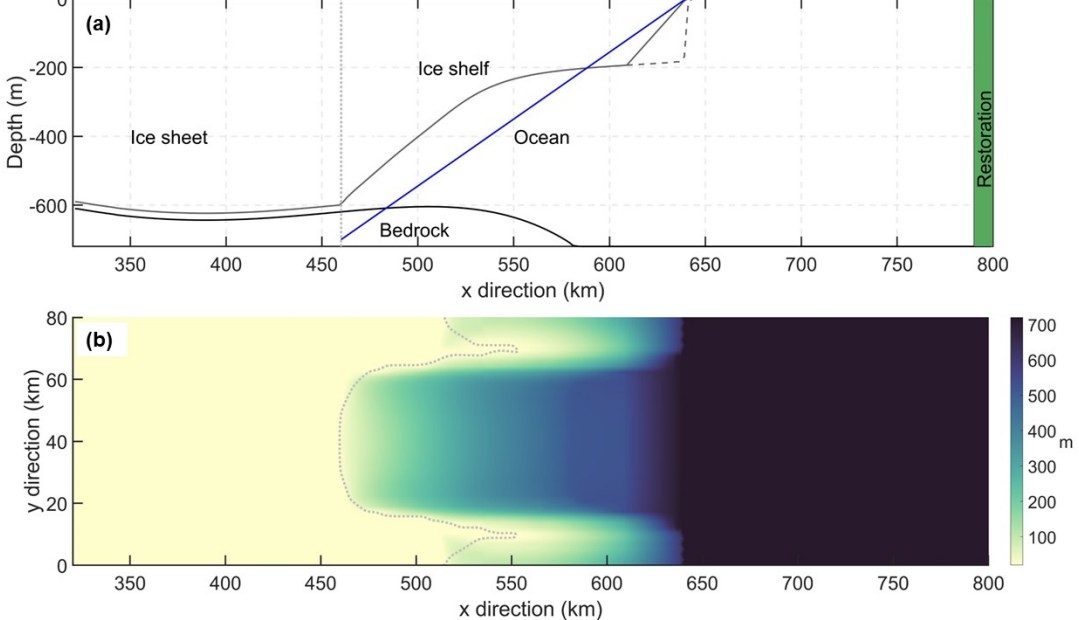

**Figure 2.** Panel (a) shows a cross-sectional view along the center of the ISOMIP+ domain, highlighting key components: the bottom topography (black line), the initial grounding line position (light gray dashed line), and the ice shelf geometry used in the FVCOM-ISOMIP+ simulations (dark grey line). Note that the ROMS-ISOMIP+ configuration uses the same ice cavity geometry except at the ice front (610 km $\leq x \leq$ 640 km), and the grey dashed lines indicate the ice front in this configuration. The blue line indicates the ice shelf geometry used in the FVCOM-Wedge simulations. The green shaded area marks the region of the ocean forcing restoration. Panel (b) displays a plane view of the ISOMIP+ domain, with color shading indicating the water column thickness. The light grey dashed lines denote the initial boundary separating the wet cells (to the right) and the dry cells (to the left). This ISOMIP+ domain also serves as the domain for the ocean component in the coupled ice sheet-ocean model experiments.

(2016). These profiles represent typical ocean conditions near Antarctic ice shelves, with "COLD" and "WARM" referring to
the conditions near cold and warm ice shelf cavities, respectively. The MEAN profiles, derived by averaging the COLD and WARM profiles, qualitatively represent average ocean properties.

The oscillating profiles are conducted as repeating cosine waves, fluctuating between the COLD and WARM profiles with a period P as

$$T_P(t) = 0.5(T_W + T_C) - 0.5(T_W - T_C)cos(\frac{2\pi}{P}t), \tag{7}$$

and

$$S_P(t) = 0.5(S_W + S_C) - 0.5(S_W - S_C)cos(\frac{2\pi}{P}t). \tag{8}$$

**Table 2.** Summary of parameters for the temperature and salinity profiles. Note that all salinities are on the practical salinity scale (PSS-78).

| Profiles | Surface temperature, $T_0$ | Bottom temperature, $T_b$ | Surface salinity, $S_0$ | Bottom salinity, $S_b$ |
|---|---|---|---|---|
| COLD | -1.9°C | -1.9°C | 33.8 | 34.55 |
| MEAN | -1.9°C | -0.45°C | 33.8 | 34.625 |
| WARM | -1.9°C | 1°C | 33.8 | 34.7 |

Here $T_P$ and $S_P$ stand for the oscillating profiles for potential temperature and salinity, respectively. $T_W$ and $S_W$ are the linear WARM profiles for potential temperature and salinity, respectively, and $T_C$ and $S_C$ are the linear COLD profiles for potential temperature and salinity, respectively. When averaged over the period P, these oscillating profiles yield the MEAN profiles.

Table 3 summarizes the stand-alone ocean model experiments. For each configuration, we conduct three constant forcing simulations and a number of oscillating forcing simulations with different periods. Specifically, oscillation periods for FVCOM-ISOMIP+ are 0.1, 0.2, 0.6, 1, 2, 6, 10, 20, and 30 years. Periods for FVCOM-Wedge are 0.1, 0.2, 1, 2, 10, and 20 years. Periods for ROMS-ISOMIP+ are 0.4, 4, 8, 20, and 36 years. While all the constant forcing simulations are initialized from the COLD rest-state cavity, the oscillating simulations are initialized from the spun-up state of the respective COLD

forcing simulations. Each simulation was run until a quasi-equilibrium state was achieved, characterized as a constant state of mean melting for the constant forcing simulations and a repetitive state for the oscillating forcing simulations. Here the quasi-equilibrium state refers to the model's spun-up phase, in which the model's outputs are no longer influenced by the initial conditions but are instead determined by the external forcings. Unless stated otherwise, our analysis is based on results from a quasi-equilibrium state, which are time-averaged over the final year of model time for constant forcing simulations and over

the last cycle for oscillating forcing simulations.

## 3.2   Melting response to oscillating ocean forcing

As Since we will explore the melting response to oscillating ocean forcing in comparison with to the constant MEAN forcing, it is necessary first to examine the melting response from the simulations restored to the MEAN profiles. Throughout the text, all the measures derived from these MEAN forcing simulations are referred to as mean-state measures. Despite their

different cavity geometries, the two FVCOM-based MEAN forcing simulations exhibit similar barotropic circulation patterns (Figure 3a and c). Both simulations display a single clockwise gyre in the open ocean. Within the ice cavity, the circulation primarily exhibits a geostrophically controlled flow, featuring an inflow of boundary waters across the ice front and along the lower flank and an outflow along the upper flank. Consequently, in both FVCOM-based simulations, intense melting is observed near the deepest part of the lower flank, while significant freezing occurs along the upper flank (Figure 3b and d). However, the

strength of the circulation and the associated melting-freezing process varies with the different ice cavity geometries. Notably, the simulation with the ISOMIP+ cavity (FI_C2M; Figure 3a) exhibits much weaker circulation compared to that with the

**Table 3.** Summary of stand-alone ocean model experiments.

| Experiment class | Simulation name | Initial state | Restoring forcing profiles |
|---|---|---|---|
| FVCOM-ISOMIP+ | FI_C2C | at rest, COLD | COLD |
| | FI_C2M | at rest, COLD | MEAN |
| | FI_C2W | at rest, COLD | WARM |
| | FI_P | FI_C2C spun-up state | oscillating, period P |
| FVCOM-Wedge | FW_C2C | at rest, COLD | COLD |
| | FW_C2M | at rest, COLD | MEAN |
| | FW_C2W | at rest, COLD | WARM |
| | FW_P | FW_C2C spun-up state | oscillating, period P |
| ROMS-ISOMIP+ | RI_C2C | at rest, COLD | COLD |
| | RI_C2M | at rest, COLD | MEAN |
| | RI_C2W | at rest, COLD | WARM |
| | RI_P | RI_C2C spun-up state | oscillating, period P |

wedge-shaped cavity (FW_C2M; Figure 3c), highlighting the effect of cavity geometry on the circulation and, consequently, on melting patterns.

In contrast, despite using the same ISOMIP+ cavity and being restored to the same MEAN profiles, the ROMS-based
MEAN forcing simulation exhibits distinct barotropic circulation patterns (Figure 3e) compared to its ~~FVCOM~~ FVCOM-based counterpart (Figure 3a). Specifically, it features three gyres in the open ocean and an inflow across the lower part of the ice front ($y = 0 - 20$ km), along with an anti-clockwise gyre near the ice front in the cavity. Additionally, basal melting in the ROMS-based simulation (Figure 3f) is generally weaker than that in the FVCOM-based simulation (Figure 3b). It is important to note that our focus here is to understand the melting response within each model configuration to oscillating ocean forcing,
rather than directly comparing the two models. The observed differences between the ROMS-based and the FVCOM-based ISOMIP+ simulations may reflect a combination of differences in model numerics (Table 1) and artifacts associated with the pressure gradient error (Zhou and Hattermann, 2020). In the FVCOM-ISOMIP+ configuration, the ice front was smoothed to reduce the pressure gradient error, a step not taken in the ROMS-ISOMIP+ configuration, which likely resulted in a smaller pressure gradient error in the FVCOM-ISOMIP+ simulations.
Table 4 presents cavity-averaged melt rates alongside the mean-state cavity residence time (MCRT) from these three MEAN forcing simulations. The MCRT, computed by dividing the cavity volume by the cavity-averaged barotropic streamfunctions, represents the time required for all the cavity waters to be flushed with the MEAN forcing waters to fully affect the basal melting (Holland, 2017). Next, we will use the MCRT as a key timescale to investigate the response of basal melting to oscillating ocean forcing.

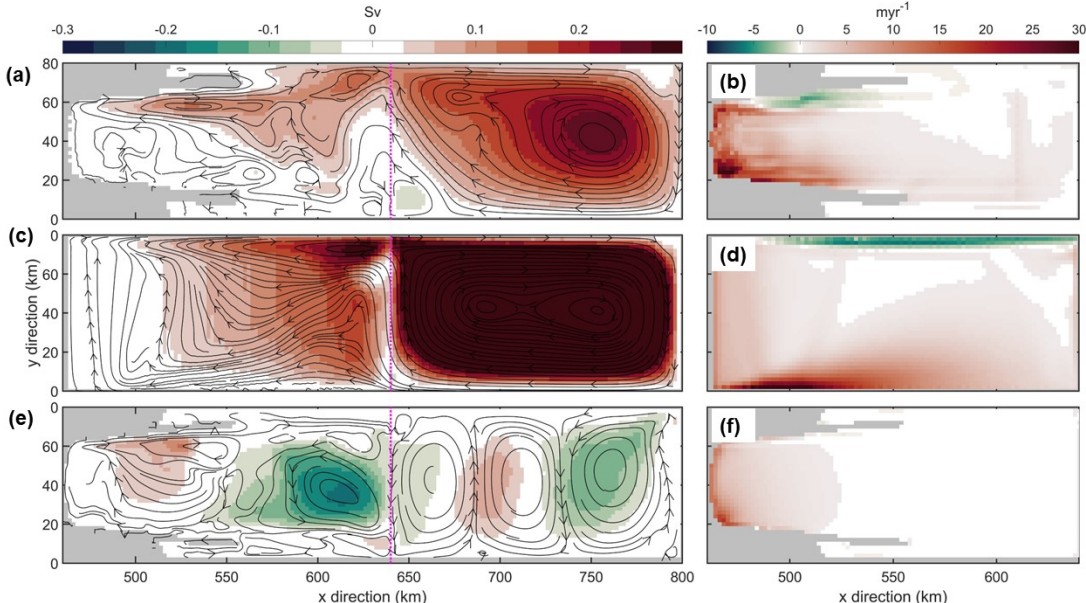

**Figure 3.** Panels (a), (c), and (e) show plane views of the quasi-steady state barotropic streamfunction from the simulations restored to the MEAN profiles for the FVCOM-ISOMIP+ (FI_C2M), FVCOM-Wedge (FW_C2M), and ROMS-ISOMIP+ (RI_C2M) configurations, respectively. The black lines and arrows indicate the barotropic flow, while the magenta dashed lines mark the ice front. Panels (b), (d), and (f) display spatial distributions of melt rates in FI_C2M, FW_C2M, and RI_C2M, respectively.

**Table 4.** Melting diagnostics for simulations restored to the MEAN profiles across different model configurations.

| Simulation name | Cavity volume ($m^3$) | Cavity-averaged barotropic streamfunction (Sv) | Cavity-averaged melt rate ($myr^{-1}$) | Mean cavity residence time (yr) |
|---|---|---|---|---|
| FI_C2M | $3.9 \times 10^{12}$ | 0.03 | 1.87 | $\sim 4$ |
| FW_C2M | $4.6 \times 10^{12}$ | 0.07 | 2.7 | $\sim 2$ |
| RI_C2M | $3.75 \times 10^{12}$ | 0.018 | 0.97 | $\sim 7$ |

Figure 4 displays the time series of domain-averaged temperatures and cavity-averaged melt rates from selected simulations for each of the three model configurations: three constant forcing simulations (COLD, MEAN, and WARM) and three oscillating simulations with periods that are either shorter than 0.1 times the MCRT, close to or within 2 times the MCRT, or longer than 5 times the MCRT. Across the three model configurations, the WARM forcing simulation displays the highest mean temperatures and, consequently, the highest mean melt rates. Notably, although the mean-state temperature is interme-
diate between the WARM and COLD forcing simulations, the corresponding mean-state melt rate stays more closely with the

COLD forcing simulation rather than evenly between the two. This suggests a non-linear, possibly quadratic relation between ocean temperatures and melt rates (Holland et al., 2008b; Jenkins et al., 2018).

In all the oscillating forcing simulations, the time series of domain-averaged temperatures and cavity-averaged melt rates exhibit oscillation patterns that reflect the periods of the respective ocean forcing. Additionally, the time series of cavity-averaged melt rates from the longer-period simulations (orange lines in Figure 4d,e, ~~and~~ f) exhibit a distinct asymmetrical shape, characterized by broader low melt troughs and narrower high melt peaks. The asymmetry is related to the internal feedback between cavity circulation and boundary forcing (Holland, 2017). During a ~~forcing~~ cycle where the ~~far-field ocean conditions vary~~ ocean forcing varies from WARM to COLD and back to WARM, when the cavity is filled with WARM water, the enhanced melting leads to faster cavity circulation, facilitating quicker ~~flushing of COLD water~~ COLD water flushing and rapid cooling of the cavity. Conversely, in a COLD cavity state, the circulation slows, extending the time taken to flush WARM water into the cavity, hence resulting in a slower warming phase. The asymmetry is also visible in the melt rate time series from the simulations with forcing periods close to the MCRT (cyan lines in Figure 4d,e, ~~and~~ f).

Furthermore, the temporal melting responses in the oscillating forcing simulations share ~~common~~ several features across the three model configurations (Figure 4d, e, and f): i) for periods significantly shorter than the MCRT, indicated by green lines, melt rates are slightly above their respective mean-state values; ii) for periods close to the MCRT, shown by cyan lines, most of the melt rates within each cycle fall below the mean-state values, suggesting reduced melting in these simulations; iii) for periods substantially longer than the MCRT, as depicted by orange lines, the maximum (minimum) melt rates are close to or equal to those from the respective WARM (COLD) forcing simulations, with a greater portion of the melt rates above the mean-state value in each cycle.

These features are also evident in the spatial distributions of melt rate deviations in the selected oscillating forcing simulations across the three model configurations ~~,~~ relative to their respective mean-state melt rates (Figure 5). For periods significantly shorter than the MCRT (top row panels in Figure 5), enhanced melting is mainly observed in the inner part of the cavity, with the cavity-averaged melt rates increasing by 10%, 5%, and 14% for the FVCOM-ISOMIP+, FVCOM-Wedge, and ROMS-ISOMIP+ simulations, respectively. For periods close to the MCRT (middle row panels in Figure 5), a significant reduction in melting occurs at locations of strong mean-state melting, with the cavity-averaged melt rates decreasing by 31%, 29%, and 29%, respectively. For periods substantially longer than the MCRT (bottom row panels in Figure 5), there is a general increase in melting, particularly in the inner part of the cavity in the two FVCOM-based simulations (Figure 5g ~~,~~ and h), with the cavity-averaged melt rates rising by 29%, 12%, and 13%, respectively.

Figure 6 provides a qualitative summary of melting response to oscillating forcing across all the simulations in the three model configurations, by depicting the relationship between normalized melt rate and normalized timescale. The normalized melt rate is computed by dividing the cavity-averaged melt rate of each oscillating forcing simulation by the corresponding mean-state cavity-averaged melt rate. Similarly, the normalized timescale is the ratio of the oscillating period to the respective MCRT. Since we use Log2(Normalized timescale) for the x-axis in the figure, a value of 0 indicates a forcing oscillation period ~~close to the~~ the same as MCRT, a value of -2 indicates a forcing period of 0.25 times the MCRT, ~~while~~ and a value of 2 indicates a forcing period of 4 times the MCRT.

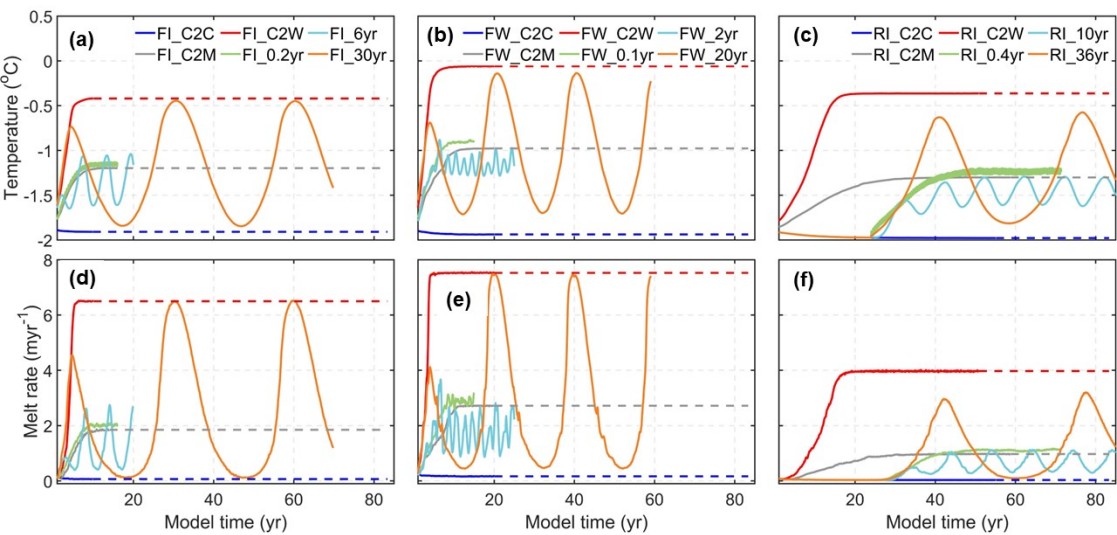

**Figure 4.** Panels (a) and (d) show time series of domain-averaged temperatures and cavity-averaged melt rates, respectively, from the se-lected FVCOM-ISOMIP+ simulations. These includes three constant forcing simulations: COLD (FI_C2C), MEAN (FI_C2M), and WARM (FI_C2W), as well as three oscillating forcing simulations with periods of 0.2 (FI_0.2yr), 6 (FI_6yr), and 30 years (FI_30yr). Panels (b) and (e) display time series of domain-averaged temperatures and cavity-averaged melt rates from the selected FVCOM-Wedge simulations, respectively. These include three constant forcing simulations: COLD (FW_C2C), MEAN (FW_C2M), and WARM (FW_C2W), as well as three oscillating forcing simulations with periods of 0.1 (FW_0.1yr), 2 (FW_2yr), and 20 years (FW_20yr). Panels (c) and (f) show time series of domain-averaged temperatures and cavity-averaged melt rates from the selected ROMS-ISOMIP+ simulations, respectively. These include three constant forcing simulations: COLD (RI_C2C), MEAN (RI_C2M), and WARM (RI_C2W), as well as three oscillating forcing simulations with periods of 0.4 (RI_0.4yr), 10 (RI_10yr), and 36 years (RI_36yr). The dashed lines extend from their respective quasi-steady state values for interpretative purposes.

Figure 6 not only reinforces the three distinct melting regimes observed in the time series and spatial distribution figures but also provides additional insights for predetermining suitable scenarios for the accelerated forcing approach. First, the normalized melt rates reach their minimum across all three model configurations when the oscillation periods approximate the MCRTs (Log2(Normalized timescale) $\approx$ 0). In this regime, melt-induced circulation begins to increase with the warm phase of the oscillation just as the forcing shifts back to the cold phase, which rapidly cools the cavity. The return to the warm phase is slower due to diminished melt-induced circulation in the cold phase, resulting in a cavity temperature closer to the COLD profiles, thereby minimizing melting. This suggests that with any adjustments in the timescale when the oscillation period approximates the MCRT, the melting response is likely to deviate significantly from the mean melting response. Secondly, when oscillation periods are shorter than the MCRT (Log2(Normalized timescale) < 0), the melting rates tend to stabilize, as indicated by normalized melt rates clustering between 0.9 and 1.1. In this regime, where the ocean ~~conditions oscillate~~ forcing

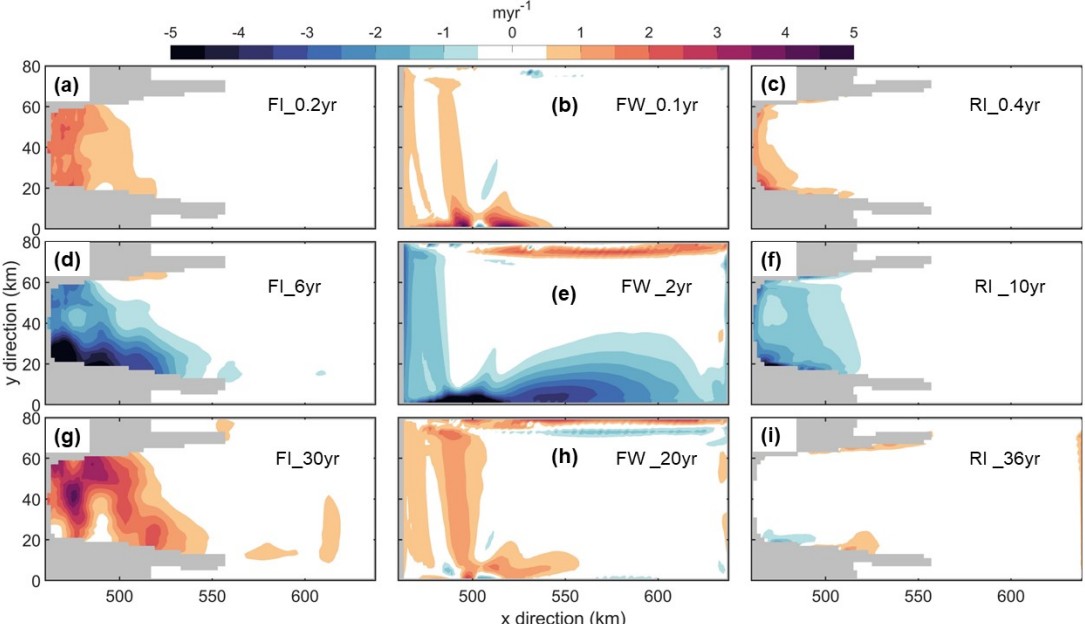

**Figure 5.** Panels (a), (d), and (g) show spatial distributions of melt rate deviations from the FVCOM-ISOMIP+ oscillating forcing simulations with periods of 0.2 (FI_0.2yr), 6 (FI_6yr), and 30 years (FI_30yr), respectively, relative to the mean-state melt rates. Panels (b), (e), and (h) display spatial distributions of melt rate deviations from the FVCOM-Wedge oscillating forcing simulations with periods of 0.1 (FW_0.1yr), 2 (FW_2yr), and 20 years (FW_20yr), respectively, relative to the mean-state melt rates. Panels (c), (f), and (i) show spatial distributions of melt rate deviations from the ROMS-ISOMIP+ oscillating forcing simulations with periods of 0.4 (RI_0.4yr), 10 (RI_10yr), and 36 years (RI_36yr), respectively, relative to the mean-state melt rates.

oscillates rapidly, the ocean temperature doesn't have time to adjust to that of the WARM or COLD profiles. This results in the water entering the cavity at a temperature close to that of the MEAN profiles, thereby leading to a melting response ~~that is~~ nearly equivalent to that observed under the MEAN forcing. Consequently, this response exhibits low sensitivity to rapidly varying ocean forcing. In contrast, melt rates increase significantly when the oscillating forcing periods greatly exceed the

MCRTs~~, melt rates increase significantly~~. Specifically, the normalized melt rates increase from about 0.7 when the forcing period near the MCRT (Log2(Normalized timescale) $\approx$ 0) to more than 1.1 when the period is much longer than the MCRT (Log2(Normalized timescale) > 2 ) for both the ISOMIP+ domain configurations. In the FVCOM-ISOMIP+ configuration, the normalized melt rate increases to about 1.3 when the forcing period (30 years) is seven times longer than the MCRT of 4 years. In addition, the FVCOM-Wedge simulations display a comparable increasing trend but at a slower rate, likely due to

differences in cavity geometry. The increase in melt rates is attributed to the quadratic relationship between melt rates and ocean temperatures (Holland et al., 2008a; Jenkins et al., 2018). In detail, as the ocean forcing oscillates slowly, ocean temperatures tend to follow the oscillatory forcing at every stage. When averaged over the oscillation period, the mean melt rate aligns more closely with that from the WARM forcing and thus is higher than that from the MEAN forcing. We expect the melting response

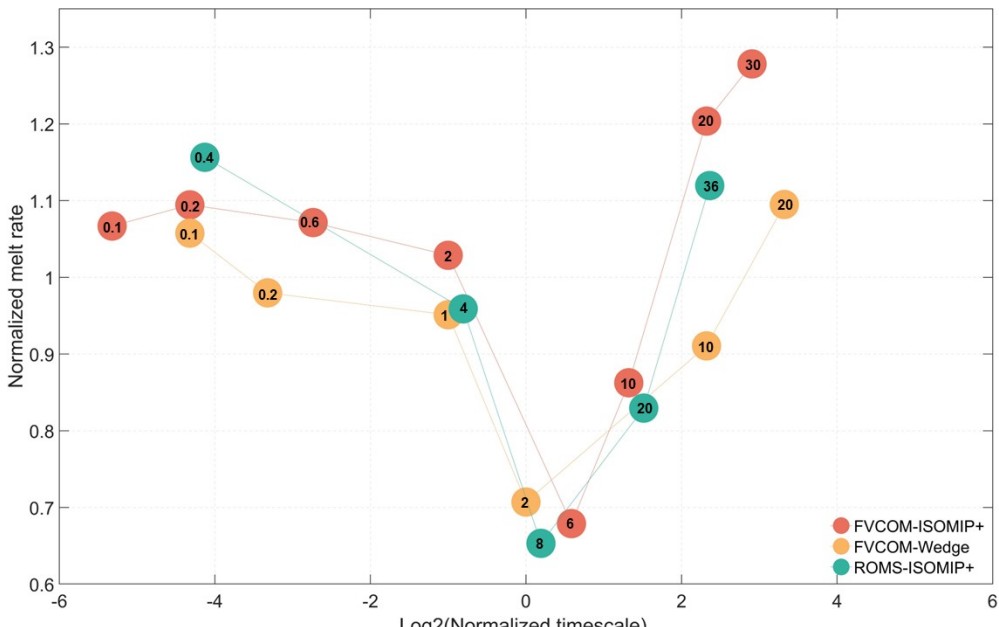

**Figure 6.** Normalized melt rates plotted against normalized timescales from all the simulations across the FVCOM-ISOMIP+, FVCOM-Wedge, and ROMS-ISOMIP+ model configurations. Numbers in the colored circles denote the period of oscillating profiles used in each simulation.

to stabilize when ocean temperatures fully adjust to the oscillatory forcing. However, ~~due to the lack of simulations with longer~~
~~periods, we are unable to~~ we cannot determine the minimum period necessary for the melting response to stabilize due to the
lack of simulations with longer periods.

In summary, the accelerated forcing approach is likely appropriate ~~in scenarios~~ when the forcing period is either significantly shorter than the MCRT or long enough to allow the melting response to fully adjust to the ocean forcing. In these cases, changes in the forcing timescale lead to a similar mean melting response. However, our approach may not be suitable when the forcing
period, whether accelerated or not, is such that any changes in the forcing timescale likely lead to deviations from the mean melting response. In the following section, we will use coupled ice sheet-ocean model experiments to verify these findings and evaluate the accelerated forcing approach.

## 4 Coupled ice sheet-ocean model experiments

### 4.1 Experiment design

To explore the approach's applicability across various ocean models, we conduct our main experiments using two coupled model setups: Elmer/Ice-FVCOM and Elmer/Ice-ROMS.

**Table 5.** Summary of coupled ice sheet-ocean model experiments. For the Constant class, "Rest" in the simulation name indicates the simulation is initialized from the COLD rest-state cavity, while "Spunup" indicates the simulation is initialized from the quasi-steady state of the respective regular forcing simulation (acceleration factor 1). The optional suffix "_R" in the simulation name denotes the use of ROMS in the coupled model. The period shown in brackets refers to model time not represented time (which is always 0.6 yr for fast and 30 yr for slow periodic forcing).

| Experiment class | Simulation name | Acceleration factor | Restoring forcing profiles (period) |
|---|---|---|---|
| Constant | CRest1(_R) | 1 | WARM |
| | CRest3(_R) | 3 | WARM |
| | CRest10(_R) | 10 | WARM |
| | CSpunup3(_R) | 3 | WARM |
| | CSpunup10(_R) | 10 | WARM |
| Periodic-fast | PFast1 | 1 | oscillating (0.6 yr) |
| | PFast3 | 3 | oscillating (0.2 yr) |
| | PFast10 | 10 | oscillating (0.06 yr) |
| Periodic-slow | PSlow1 | 1 | oscillating (30 yr) |
| | PSlow1.5 | 1.5 | oscillating (20 yr) |
| | PSlow3 | 3 | oscillating (10 yr) |

Our coupled model experiments are based on the MISOMIP1 IceOcean1 experiment framework (Asay-Davis et al., 2016). The ocean model domain is identical to the ISOMIP+ domain used in the stand-alone ocean model experiments (Figure 2), while the ice sheet model domain extends from 0 to 640 km in the x direction. We have structured the coupled model experi-

ments into three classes characterized by the timescale of the restored ocean forcing: Constant, Periodic-fast, and Periodic-slow. Each class includes one benchmark simulation under regular forcing and several simulations under accelerated forcing, as listed in Table 5 and explained in detail below.

The Constant class represents a scenario where an ice shelf cavity experiences a regime shift from a cold to a warm cavity. Each coupled model setup has one regular forcing and four accelerated forcing simulations. The regular forcing simulation,

identical to the COLD-to-WARM MISOMIP1 IceOcean1r experiment (Asay-Davis et al., 2016), is initialized from the COLD rest-state cavity and restored to the constant WARM profiles. Two accelerated forcing simulations, using acceleration factors of 3 and 10, are initialized from the COLD rest-state cavity. Another two accelerated forcing simulations, with the same acceleration factors, are initialized from the spun-up state of the respective regular forcing simulation. In detail, the FVCOM-based and ROMS-based simulations are initialized from the model state at 12 years and 20 years, respectively, of the corresponding

regular forcing simulation. All accelerated forcing simulations are restored to the same WARM profiles as the regular forcing

simulation. This is because the timescale of a constant forcing can be considered infinite, and any accelerations of it are also infinite. We run simulations for 100 years in represented time. Note that we use both Elmer/Ice-FVCOM and Elmer/Ice-ROMS for this class. However, we only use Elmer/Ice-FVCOM for the following two classes due to resource constraints.

The Periodic-fast class represents a scenario where an ice shelf cavity experiences fast-varying far-field ocean conditions with a timescale much shorter than its cavity residence time. The regular forcing simulation is restored to the oscillating profiles with a period of 0.6 years, ~~which is~~ significantly shorter than the MCRT of 4 years. Two accelerated forcing simulations, using acceleration factors of 3 and 10, are restored to the oscillating profiles with periods of 0.2 and 0.06 years in model time, respectively. All simulations in this class are initialized from the model state at 12 years of the stand-alone simulation with the MEAN profiles (FI_C2M). We run all simulations for 30 years in represented time, beyond their spin-up phase and reaching a quasi-equilibrium state.

The Periodic-slow class represents a scenario where an ice shelf cavity experiences slow-varying far-field ocean conditions that vary over a timescale significantly longer than its cavity residence time. The regular forcing simulation is restored to the oscillating profiles with a period of 30 years, more than seven times longer than the MCRT of 4 years. Longer periods are not feasible in the current configuration due to computational constraints. Two accelerated forcing simulations, using acceleration factors of 1.5 and 3, are restored to the oscillating profiles with periods of 20 and 10 years in model time, respectively. All simulations in this class are initialized from the 50-year spun-up state of the stand-alone FVCOM-ISOMIP+ oscillating forcing simulation with a period of 30 years. We run all simulations for 55 years in represented time.

## 4.2 Evaluating the accelerated forcing approach

We now evaluate the accelerated forcing approach by directly comparing key diagnostics relevant to basal melting from the accelerated forcing simulations to those from the regular forcing simulation within each of the three experiment classes. These diagnostics include time series of cavity-averaged melt rates and ocean volume changes, along with spatial distributions of melt rates, and integrated ice draft changes at the end of the simulation. Note that the spatial distributions of melt rates are time-averaged over the last year of represented time for the Constant class, the last 6 years of represented time for the Periodic-fast class, and over the last cycle of represented time for the Periodic-slow class. Additionally, given that the absolute differences in ocean-driven melting under ~~the~~ accelerated forcing are concentrated near the grounding line across all experiment classes—a detail that will be elaborated on below—and considering that marine ice sheets are sensitive to melt patterns near the grounding line, we also evaluate grounding line positions throughout the simulation to assess the net effect of melting differences on ice dynamics under accelerated forcing.

### 4.2.1 Constant ocean forcing

When the coupled model is initialized from the COLD rest-state cavity and undergoes the WARM forcing, the FVCOM-based simulations take a similar time (about 11 years) to adjust to the transient forcing changes, whether under regular forcing or accelerated forcing (Figure 7a). During this adjustment, warmer water from the boundaries flushes into the ice cavity, causing melt rates to rise from 0 to $\sim 2.4 \ \mathrm{myr}^{-1}$ in all simulations ~~,~~ despite the different accelerated ice draft change rates. This pattern

indicates that the spin-up duration is mainly dictated by ocean boundary conditions, not by the feedback in the coupled model system. Consequently, the melting response during the spin-up phase in the regular forcing simulation is not reproduced in the accelerated forcing simulations when viewed in represented time (CRest3 ~~,~~ and CRest10; Figure 7b), suggesting the accelerated forcing approach is not applicable during this phase. This conclusion is supported by results from the ROMS-based simulations (Figure 7b ~~,~~ and d).

In contrast, the accelerated forcing simulations initialized from a spun-up state exhibit a similar temporal melting response to the regular forcing simulation. For the FVCOM-based setup, both accelerated forcing simulations yield a constant melt rate of about $2.4 \text{ myr}^{-1}$ (CSpunup3 ~~,~~ and CSpunup10; Figure 7b), nearly identical to that in the regular forcing simulation. For the ROMS-based setup, the accelerated forcing approach can effectively capture oscillations in the melt rates (Figure 7d), which are primarily attributed to an ocean response to the ice draft changes (Zhao et al., 2022), with comparable frequency and magnitude to those under the regular forcing. Thus, our subsequent analysis focuses only on the accelerated forcing simulations initialized from the spun-up state.

Basal melting in the FVCOM-based accelerated forcing simulations ~~exhibit~~ exhibits a similar spatial pattern as the regular forcing simulation, with minor exceptions in the deeper parts of the cavity. ~~In~~ Under the regular forcing ~~simulation~~ (Figure 8a), enhanced melting is observed in the deep ice region near the grounding line (x< 450 km) with a region-averaged melt rate of $24 \text{ myr}^{-1}$. Absolute differences in melt rates in both accelerated forcing simulations relative to the regular forcing simulation are lower than $0.5 \text{ myr}^{-1}$ across most of the cavity, indicated by the large uncolored areas away from the deep ice region in Figure 8c and e. In the deep ice region, averaged melt rate differences relative to the regular forcing simulation are $1.5 \text{ myr}^{-1}$ and $3 \text{ myr}^{-1}$ for the simulations with acceleration factors of 3 and 10, respectively, corresponding to changes of approximately 6% and 12%.

A similar conclusion can be drawn from the ROMS-based coupled simulations, particularly for the lower acceleration factor, as shown in the right column of Figure 8. In specific, visible differences in melt rates in both accelerated forcing simulations, relative to the regular forcing simulation, mainly occur in the newly ungrounded high-melting region (x < 440 km; Figure 8d and f). In this region, the accelerated forcing simulation with a factor of 3 shows an absolute difference in region-averaged melt rate of $5 \text{ myr}^{-1}$, representing a relative change of 6% of the region-averaged melt rate of $79 \text{ myr}^{-1}$ in the regular forcing simulation (Figure 8b). However, the absolute difference is $43 \text{ myr}^{-1}$ in the simulation with an acceleration factor of 10, amounting to a relative change in region-averaged melting beyond 50%, likely resulting from a phase shift in the melt rate oscillation.

Figure 9 displays the spatial distributions of integrated ice draft changes and the differences in ice draft changes between the accelerated and the regular forcing simulations for both FVCOM-based and ROMS-based setups. For the FVCOM-based setup, the integrated ice draft changes under regular forcing increase from about 50 m at the ice front to approximately 350 m near the original grounding line (x $\approx$ 460 km) and then decreases to below 50 m in the newly-ungrounded area (Figure 9a). The absolute differences in ice draft changes are typically below 5 m under the accelerated forcing with a factor of 3 and below 10 m for a factor of 10 across most of the cavity (Figure 9c and e ). However, in some ~~area~~ areas near the grounding line, these differences increase to over 10 m and 20 m for factors of 3 and 10, respectively, representing relative changes exceeding 20%

and 40%. For the ROMS-based setup, integrated ice draft changes under regular forcing increase from about 50 m at the ice front to over 400 m at approximately 15 km from the grounding line and then decrease to about 100 m at the grounding line (Figure 9b). Significant deviations in the ice draft changes under accelerated forcing, relative to the regular forcing simulation, are concentrated within 15 km of the grounding line (Figure 9 d and f). In some areas within this region, the absolute differences exceed 20 m and 50 m under accelerated forcing with factors of 3 and 10, respectively, corresponding to relative changes of more than 20% and 50%.

Ice draft changes lead to ocean volume changes in the coupled system. In both the FVCOM-based and ROMS-based simulations, the time series of ocean volume changes are nearly identical to those in the corresponding regular forcing simulations. They show a steady increase from 0 to approximately $2\times10^{12}$ m$^3$ and $1.7\times10^{12}$ m$^3$, respectively, over 100-year period in represented time (Figure 10a and b ). Furthermore, the time series of grounding line positions along the central line (y = 40 km) show identical grounding line retreat under regular and accelerated forcing with different factors for the FVCOM-based setup (Figure 10c), with only minor variations in the timing of ungrounding of individual model grid elements. A similar conclusion can be drawn from the ROMS-based simulations, except for the accelerated forcing simulation with a factor of 10, likely due to the phase shift in melt rate oscillations.

Our findings, as described above, suggest that the accelerated forcing approach is applicable in scenarios where an ice shelf cavity experiences steady far-field ocean conditions. This is particularly true for the lower acceleration factor, evidenced by the less than 10% relative changes in integrated ice draft changes across most of the cavity, less than 6% in melt rates, and nearly identical ocean volume changes and grounding line retreat in the case of an acceleration factor of 3 in both FVCOM-based and ROMS-based setups. However, given that the temporal melting response in the accelerated forcing simulations diverges from the regular forcing simulation during the spin-up phase, the accelerated forcing approach only applies when the system is in the spun-up phase.

### 4.2.2  Fast-varying ocean forcing

When the coupled model is restored to fast-varying ocean forcing with a timescale much shorter than the MCRT, the regular forcing simulation shows high-frequency variability in temporal melting response (gray lines in Figure 11a). This variability is not captured in the accelerated forcing simulations (red and orange lines in Figure 11a). However, for the evolution of the coupled ice-ocean system, the time-averaged melting response is more important than the high-frequency variability. The accelerated forcing simulations exhibit melt rates that fluctuate around the low-pass filtered melt rate time series of the regular forcing simulation (black lines in Figure 11a). The maximum deviation in melt rates in the accelerated forcing simulations, relative to the regular forcing simulation, is approximately $0.1$ myr$^{-1}$, less than 10 % of the corresponding melt rate under regular forcing. This finding agrees well with the findings from the stand-alone ocean experiments presented earlier in Section 3, showing that the mean melting response to fast-varying ocean forcing converges to the mean-state melting response.

The spatial pattern of melting in the regular forcing simulation is characterized by enhanced melting near the grounding line, reaching up to $30$ myr$^{-1}$ (Figure 12a), with an asymmetric pattern of melting at the lower part and freezing at the upper part of the cavity approximately 20 km away from the grounding line. This pattern shows a general similarity to the one observed in

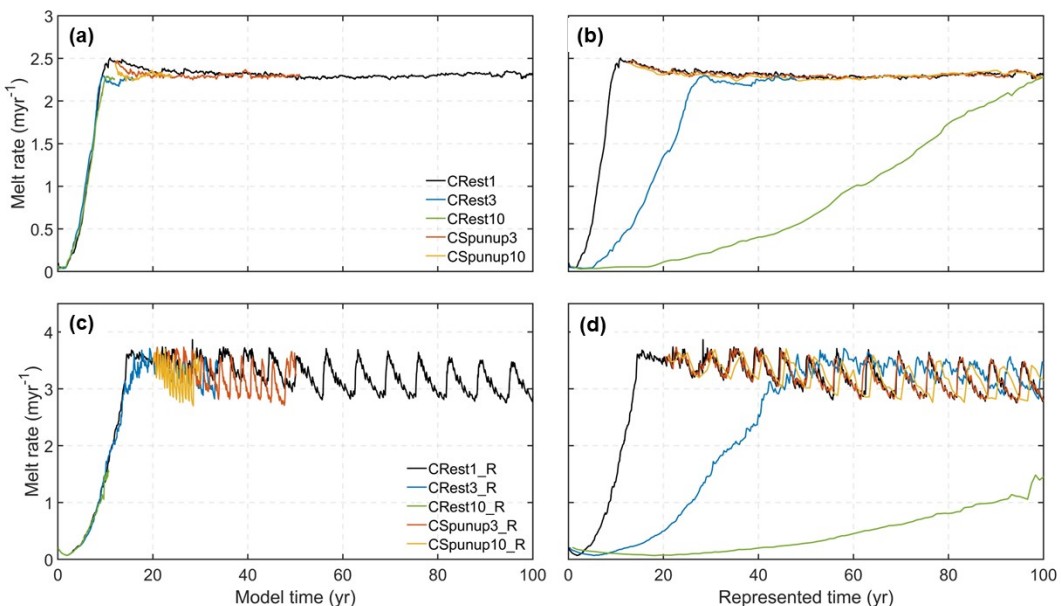

**Figure 7.** Panels (a) and (b) show time series of cavity-averaged melt rates from the FVCOM-based simulations in terms of model time and represented time, respectively. The simulations include the regular forcing simulation (CRest1), the accelerated forcing simulations initialized from the COLD rest-state cavity with factors of 3 (CRest3) and 10 (CRest10), and the accelerated forcing simulations initialized from the spun-up state with factors of 3 (CSpunup3) and 10 (CSpunup10). Panels (c) and (d) show time series of cavity-averaged melt rates from the ROMS-based simulations in terms of model time and represented time, respectively. These simulations include the regular forcing simulation (CRest1_R), the accelerated forcing simulations initialized from the COLD rest-state cavity with factors of 3 (CRest3_R) and 10 (CRest10_R), and the accelerated forcing simulations initialized from the spun-up state with factors of 3 (CSpunup3_R) and 10 (CSpunup10_R).

the stand-alone MEAN forcing simulation (FI_C2M, Figure 3b). Absolute differences in melt rates in both accelerated forcing simulations, relative to the regular forcing simulation, are lower than $0.5 \ \mathrm{myr^{-1}}$ across most of the cavity, indicated by the

large uncolored areas away from the grounding line in Figure 12c and e. However, notable deviations in melt rates are observed near the grounding line under the accelerated forcing with a factor of 10 (Figure 12e). In this region, a few locations show melt rate differences exceeding $15 \ \mathrm{myr^{-1}}$, which is more than 50 % of the corresponding melt rate in the regular forcing simulation.

Reflecting the melting pattern, the most pronounced reductions in the integrated draft in the regular forcing simulation are observed at the lower flank of the cavity and near the grounding line, reaching up to 200 m (Figure 12b). Absolute deviations in

the integrated draft changes in the accelerated forcing simulation with a factor of 3, relative to the regular forcing simulation, are small and generally below 2 m across most of the cavity, with a few locations near the grounding line exceeding 5 m (Figure 12d). In the accelerated forcing simulation with a factor of 10, the absolute deviations ~~are~~ become larger, especially in the inner part of the cavity (x < 500 km), with some locations near the grounding line exceeding 20 m (Figure 12f). Additionally,

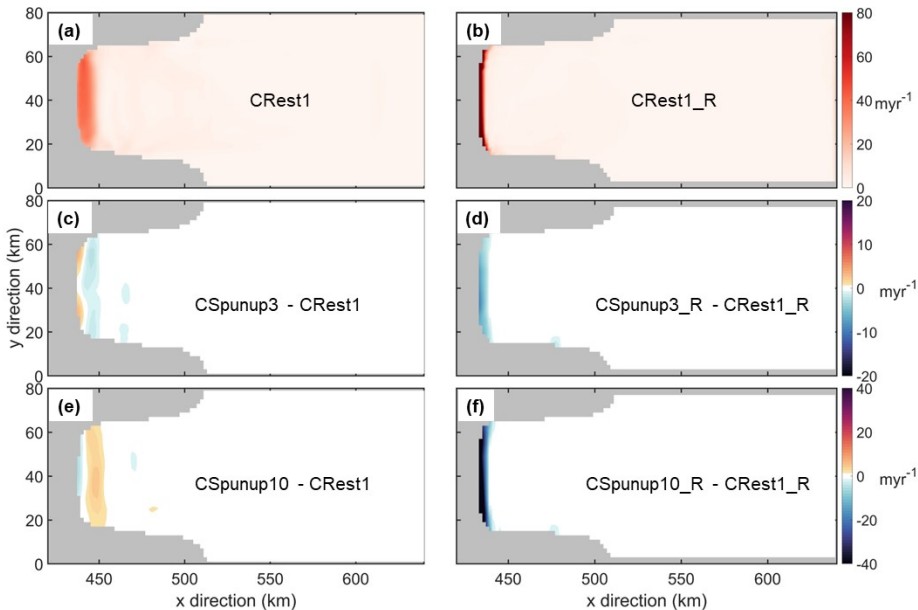

**Figure 8.** Panels (a) and (b) show melt rates ~~from~~ in the FVCOM-based regular forcing simulation (CRest1) and the ROMS-based regular forcing simulation (CRest1_R), ~~respetively~~respectively. Panels (c) and (e) display the differences in melt rates ~~from~~ in the FVCOM-based accelerated forcing simulations with factors of 3 (CSpunup3) and 10 (CSpunup10), respectively, relative to the regular forcing simulation. Panels (d) and (f) show the differences in melt rates ~~from~~ in the ROMS-based accelerated forcing simulations with factors of 3 (CSpunup3_R) and 10 (CSpunup10_R), respectively, relative to the regular forcing simulation.

the time series of the total volume changes (Figure 11b) and the grounding line positions (Figure 11c) are nearly identical over

30 years in represented time under both regular and accelerated forcing.

     In summary, our analyses indicate that the accelerated forcing simulations generally reproduce the temporal melting response, spatial distributions of melt rates, and integrated ice draft changes, with relative changes in these variables typically under 10%. However, when a higher acceleration factor is used, relative differences in melt rates and integrated ice draft changes exceed 10 % at a few locations near the grounding line. Despite these discrepancies, the total ocean volume changes

and the grounding line retreat remain identical under accelerated forcing compared to those under regular forcing. Therefore, we consider the accelerated forcing approach suitable when the forcing timescale is significantly shorter than the mean cavity residence time, as supported by our findings from the stand-alone ocean experiments.

### 4.2.3    Slow-varying ocean forcing

When the coupled model is subjected to slow-varying ocean forcing with a timescale much longer than the MCRT, the regular

forcing simulation exhibits a temporal melting response with an oscillation pattern matching the period of the ocean forcing, approximately a 30-year period in represented time (black lines in Figure 13a). Although both accelerated forcing simulations

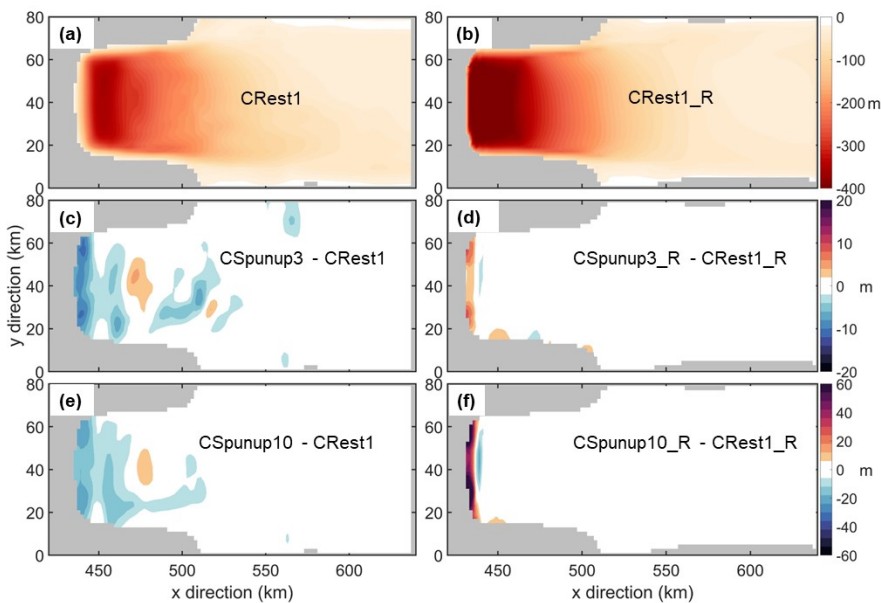

**Figure 9.** Panels (a) and (b) show integrated ice draft changes ~~from~~ in the FVCOM-based regular forcing simulation (CRest1) and the ROMS-based regular forcing simulation (CRest1_R), respectively. Panels (c) and (e) display the differences in integrated draft changes ~~from~~ in the FVCOM-based accelerated forcing simulations with factors of 3 (CSpunup3) and 10 (CSpunup10), respectively, relative to the regular forcing simulation. Panels (d) and (f) show the differences in integrated draft changes ~~from~~ in the ROMS-based accelerated forcing simulations with factors of 3 (CSpunup3_R) and 10 (CSpunup10_R), respectively, relative to the regular forcing simulation.

can capture the oscillation period, there is a noticeable reduction in the oscillation amplitude (red and orange lines in Figure 13a). Specifically, in the second cycle, peak melt rates decrease from approximately $5 \ \mathrm{myr}^{-1}$ under the regular forcing to about $4 \ \mathrm{myr}^{-1}$ under the accelerated forcing with a factor of 1.5, and to approximately $2 \ \mathrm{myr}^{-1}$ with a factor of 3. This

reduction in the melting amplitude agrees well with the findings from our stand-along experiments presented in Section 3, showing that the melting response is reduced when the ocean forcing period is not significantly different from the MCRT. In the accelerated forcing simulation with a factor of 3, the ocean forcing period is adjusted from 30 years under the regular forcing to 10 years. This adjustment brings the period closer to the MCRT of 4 years, resulting in a significant reduction in melting. In addition, the asymmetrical melt rate curve in the regular forcing simulation, characterized by a rapid rise to a peak,

a slower decline, and a prolonged period of low melt — due to internal feedback between the cavity circulation and forcing — becomes less pronounced in the accelerated forcing simulation with a factor of 1.5 and nearly disappears with a factor of 3.

The spatial melting patterns in the two accelerated forcing simulations notably differ from those in the regular forcing simulation. In the regular forcing simulation, basal melting exhibits a spatial pattern similar to the stand-alone MEAN ocean forcing simulation (FI_C2M, Figure 3b), featuring a high-melting zone exceeding $50 \ \mathrm{myr}^{-1}$ near the grounding line and an

asymmetric pattern of melting and freezing 20 km away from the grounding line (Figure 14a). However, this high-melting zone

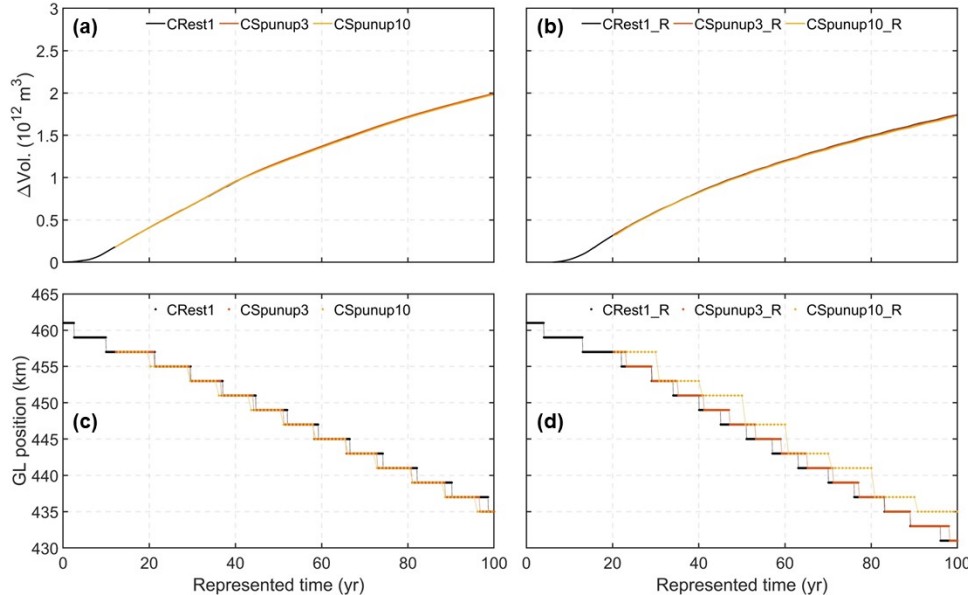

**Figure 10.** Panels (a) and (c) show time series of total ocean volume changes and grounding line ~~(GL)~~ positions, respectively, ~~from~~ in the FVCOM-based regular forcing simulation (CRest1) and the accelerated forcing simulations with factors of 3 (CSpunup3) and 10 (CSpunup10). Panels (b) and (d) display time series of total ocean volume changes and ~~GL~~ grounding line positions, respectively, ~~from~~ in the ROMS-based regular forcing simulation (CRest1_R) and the accelerated forcing simulations with factors of 3 (CSpunup3_R) and 10 (CSpunup10_R).

is less evident in both accelerated forcing simulations ( Figure 14c and e). The discrepancy is also reflected in the integrated ice draft changes: significant reductions in ice draft exceeding 300 m near the grounding line in the regular forcing simulation (Figure 14b) are only partially visible in the accelerated forcing simulation with a factor of 1.5 (Figure 14d) and nearly absent with a factor of 3 (Figure 14f).

Significant differences exist in the time series of total volume changes between the regular and accelerated forcing simulations (Figure 13b). By the end of the simulation, the total changes amount to $0.62 \times 10^{12}\mathrm{m}^3$ in the regular forcing simulation, and $0.4 \times 10^{12}\mathrm{m}^3$ and $0.18 \times 10^{12}\mathrm{m}^3$ in the accelerated forcing simulations with factors of 1.5 and 3, respectively, corresponding to relative differences of approximately 35% and 70%. Furthermore, the time series of grounding line positions (Figure 13c) also shows notable differences between the regular and the accelerated forcing simulations, with the grounding line retreating

less under accelerated forcing. By the end of the simulation, the grounding line positions are at $x = 445$ km in the regular forcing simulation and 451 km and 455 km in the accelerated forcing simulations with factors of 1.5 and 3, respectively.

In summary, the accelerated forcing simulations, particularly with the factor of 3, do not effectively replicate the melting response observed in the regular forcing simulation. This is mainly attributable to the adjusted timescale of ocean forcing in the accelerated simulation being close to the mean cavity residence time.

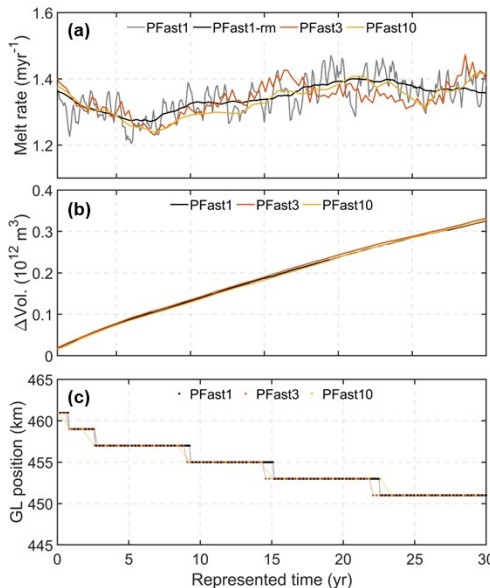

**Figure 11.** Time series of (a) cavity-averaged melt rates, (b) ocean volume changes, and (c) grounding line ~~(GL)~~ positions from the regular forcing simulation (PFast1) and the accelerated forcing simulations with factors of 3 (PFast3) and 10 (PFast10). Also shown in (a) is a low-pass filtered version of the melt rate time series from the regular forcing simulation (PFast1-rm).

## 5 Discussion and conclusions

In this study, we have introduced the accelerated forcing approach to address the discrepancy in timescales between the ice sheet and ocean components in coupled ice sheet-ocean modelling. This approach, which extends the ocean simulation duration by a constant acceleration factor, has been evaluated within the MISOMIP1 framework across three scenarios representing varied far-field ocean conditions categorized by the relative magnitude of the forcing timescale to the mean cavity residence time.

The mean cavity residence time, mainly determined by the cavity geometry and barotropic transport, is an intrinsic timescale of an ice shelf cavity. It represents the time needed for the cavity to reach an equilibrium melting state, where the cavity is filled with water that is exactly in balance with the steady ocean boundary forcing (Holland, 2017). When the timescale of the varying ocean forcing approaches this intrinsic timescale, interactions occur between basal melting, cavity circulation, heat inertia within the cavity, and transient changes in boundary forcing, leading to a melting minimum. Consequently, the melting response becomes highly sensitive to any alterations in these factors. This scenario tests the underlying assumption of the accelerated forcing approach that basal melting response is not sensitive to corresponding accelerations in ocean boundary forcing. Hence, the accelerated forcing approach likely loses applicability when the forcing timescale, whether under regular or accelerated forcing, is close to the mean cavity residence time. This limitation should be considered when applying our approach to real-world scenarios. For example, the El Niño-Southern Oscillation (ENSO), which significantly influences regions like the

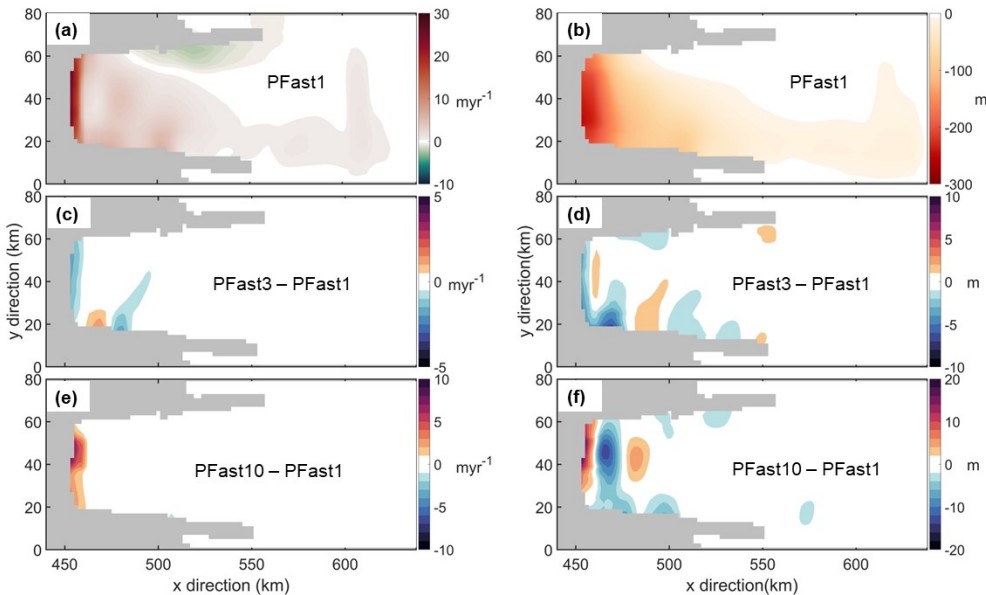

**Figure 12.** Panel (a) shows spatial distributions of melt rates from the regular forcing simulation (PFast1), while panels (c, e) display the differences in melt rates from the accelerated forcing simulations with factors of 3 (PFast3) and 10 (PFast10) relative to the regular forcing simulation. Panel (b) shows the spatial distribution of integrated ice draft changes from the regular forcing simulation (PFast1), with panels (d) and (f) showing the differences in ice draft changes from the accelerated forcing simulations with factors of 3 (PFast3) and 10 (PFast10) relative to the regular forcing simulation.

Amundsen Sea (Paolo et al., 2018; Huguenin et al., 2024), may be poorly represented under accelerated forcing due to its typical 2-7 year cycle coinciding with the cavity residence time of certain ice shelves around Antarctica. Notably, cold-water shelves with large areal extent, such as the Fichner-Ronne Ice Shelf and the Ross Ice Shelf, have a cavity residence time of 4-8 years (Nicholls and Østerhus, 2004; Loose et al., 2009). This alignment could lead to an overestimation of the melting response when ENSO's timescale is compressed under accelerated forcing. Moreover, even if the multi-decadal variation in forcing
substantially exceeds the cavity residence time, applying the accelerated forcing approach may result in an underestimation of basal melting response once its compressed timescale is comparable to the cavity residence time. Therefore, caution should be used when applying the accelerated forcing approach to studies addressing climate variability on sub-decadal to decadal timescales.

However, when the ocean forcing varies over a timescale much shorter than the cavity residence time, the ocean model
system behaves similarly to a low-pass filter. In this case, the time-average melting response is less coupled with the varying boundary forcing~~and~~. It tends to converge to a stable state produced by the time-averaged forcing, making it insensitive to changes in timescales of the forcing, as observed in the Periodic-fast experiment class. This scenario upholds the assumption

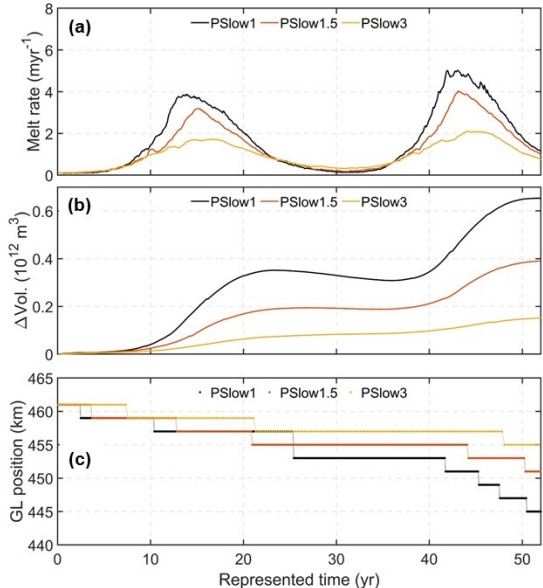

**Figure 13.** Time series of (a) cavity-averaged melt rates, (b) ocean volume changes, and (c) grounding line ~~(GL)~~ positions ~~from~~ in the regular forcing simulation (PSlow1) and the accelerated forcing simulations with factors of 1.5 (PSlow1.5) and 3 (PSlow3). The first 3 years are considered the spin-up phase and are excluded from the analysis.

of the accelerated approach. Therefore, the accelerated forcing approach can be applicable when the ocean forcing varies on seasonal timescale.

When the timescale of the ocean forcing significantly exceeds the cavity residence time, the cavity is flushed several times during each cycle. Unlike with the steady ocean forcing, the cavity can never fully achieve the equilibrium melting state under oscillating ocean forcing (Holland, 2017). Nevertheless, if the period is sufficiently long, waters at each phase of the forcing cycle may have enough time to be flushed into the cavity, allowing the melting to reach a quasi-equilibrium state. This state closely approximates equilibrium but includes slight fluctuations due to the continuous variation in forcing. For instance, a

period of 30 years seems long enough for the FVCOM-ISOMIP+ configuration to reach the quasi-equilibrium melting state at each phase of the cycle. Figure 4d illustrates that the minimum mean melt rate in the oscillating forcing with a period of 30 years (FI_30yr) deviates only slightly from that in the COLD forcing simulation, indicating that even the coldest waters have enough time to fill the cavity and influence melting. This suggests that waters in a warmer phase, particularly the warmest, also have enough time to flush into the cavity and reach a quasi-equilibrium melting state. This is supported by the maximum

mean melt rate being nearly identical to that under the WARM forcing. Considering a hypothetical 300-year forcing period, waters in each phase of the cycle would have a 10 times longer time to influence the cavity than the 30-year cycle, and the quasi-equilibrium melting state in each phase would last about 10 times longer. Therefore, the melting response in any single phase of the 300-year cycle can be approximated by the response in the corresponding single phase of the 30-year cycle, supporting the

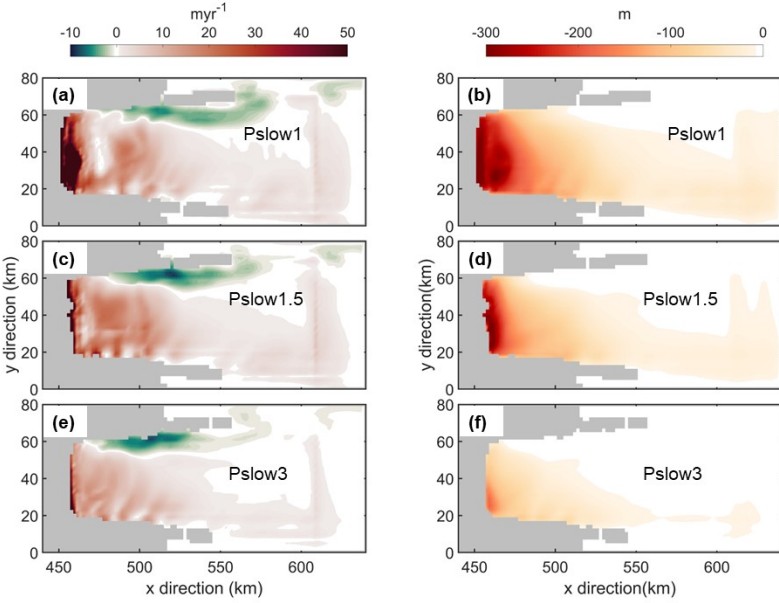

**Figure 14.** Panels (a, c, e) show spatial distributions of melt rates, while panels (b, d, f) depict spatial distributions of integrated ice draft changes. These results are derived from the regular forcing simulation (PSlow1) and the accelerated forcing simulations with factors of 1.5 (PSlow1.5) and 3 (PSlow3), respectively.

fundamental assumptions of the accelerated forcing approach. Although we have not tested forcings with periods longer than
years due to resource constraints, the constant forcing in the Constant experiment class essentially represents an infinitely slow varying force once the model reaches a quasi-steady state. This highlights the potential applicability of the accelerated forcing approach in century-long cavity-processes-oriented modelling studies, which could improve the accuracy of projections of Antarctica's contribution to sea level rise. In such projections, the slowly varying background forcing would not be periodic but instead steadily increasing at comparably slow rates in global warming scenarios. Nevertheless, the linearly increasing
trend from cold to warm can be considered a warming phase of varying forcing over even longer timescales far exceeding the mean cavity residence time of any ice shelf, ensuring the applicability of the accelerated forcing approach.

For scenarios involving mixed timescales, such as seasonal forcing superimposed on decadal oscillations with a steady background increase, additional experiments are necessary to yield definitive answers. Addressing these complex interactions requires a broader range of studies to fully understand the dynamics at play. These studies would help clarify how various
overlapping timescales influence each other, which is essential for more accurate climate modelling. Such investigations, however, fall beyond the scope of our current study and represent important directions for future research.

Nevertheless, our study demonstrates that the accelerated forcing approach can directly contribute to the MISOMIP1 project by reducing the required simulation time of 100 years, depending on the acceleration factors. Applying the accelerated approach with a factor of 3 for the IceOcean1 experiment has reduced the spun-up simulation duration by a factor of 3 and reproduced

most of the melting diagnostics within 10% of those with the regular forcing approach across two participating coupled models. Recommending the accelerated forcing approach to other participating models within the MISOMIP framework would provide a more comprehensive understanding of the robustness and applicability of the approach in idealized model setups. Furthermore, our current evaluation of the approach has been conducted using the IceOcean1 setup, where the calving front is fixed. It would be worthwhile to explore the applicability of the accelerated forcing approach using the IceOcean2 setup, which

is similar to the IceOcean1 setup but includes dynamic calving. This extension could enhance our understanding of the ocean's response to calving fluxes under accelerated forcing, which will be discussed in detail in the following paragraph.

It is important to acknowledge the limitations of our idealized study. When investigating the sensitivity of melting responses to changes in the timescale of the boundary conditions, we have only considered the lateral ocean conditions and changes in the ice draft, assuming these factors predominantly control the cavity circulation and, thus, the basal melting. This simplification

presents challenges when applied to real-world scenarios where other boundary conditions affecting the cavity properties, as well as the open ocean, can not be ignored. One of them is the total glacial meltwater input to the ocean, comprising melt due to iceberg calving, basal melting, and subglacial discharge (from the subglacial hydrologic system). Numerous studies have highlighted the significant impact of glacial meltwater on ocean stratification, with important consequences for the evolution of sea ice (Bintanja et al., 2013; Merino et al., 2018; Goldberg et al., 2023), Antarctic bottom water formation (Li et al., 2023),

ocean currents around Antarctica (Nakayama et al., 2021; Gwyther et al., 2023; Moorman et al., 2020; Bronselaer et al., 2018; Purich and England, 2023; Li et al., 2024). The current study, which focuses on fine-resolution ice sheet-ocean interactions at the Antarctic margins, specifically the ice shelf cavity, includes only the ocean-driven melt component of glacial meltwater. This is because basal meltwater has the largest impact on the cavity circulation, mainly through buoyancy forcing. Larger-scale studies would also need to quantify the impact of other components of glacial meltwater, especially the calving flux, under

accelerated forcing.

Furthermore, adjustments in glacial meltwater input are necessary to realistically represent its impacts on the ocean and climate under the accelerated forcing. Without such adjustments, the total freshwater flux into the ocean would not be consistent with that under the regular forcing, potentially distorting climate simulations. However, accelerating the meltwater flux introduces its own challenges. A significant increase in local freshwater input over a short period can drastically alter local

salinity gradients and stratification. This disruption can affect everything from mixing processes to ocean currents, potentially leading to unrealistic model behavior. Following Lofverstrom et al. (2020), we propose not accelerating the meltwater flux in order to maintain realistic local ocean dynamics. Instead, to mitigate the inconsistent freshwater input in the accelerated simulations, we suggest applying periodic restoration techniques to adjust the ocean's salinity and temperature fields using observed or targeted values (Griffies et al., 2009, 2016; Lofverstrom et al., 2020). Moreover, we expect similar inconsistencies

in atmospheric boundary conditions—such as precipitation (freshwater input), and wind and radiation fluxes (energy input)-under the accelerated forcing. The aforementioned periodic restoration techniques can also help reduce the effects of these inconsistencies, thereby ensuring more representative freshwater and energy inputs in the ocean model. For multi-centennial projections, which are the ideal target for the accelerated approach, such restoration requires prior knowledge of temperature and salinity projections. As a result, the accelerated approach is most applicable for downscaling simulations from the Coupled

Model Intercomparison Project (CMIP) using an ice sheet-ocean model, rather than for fully coupled climate models with interactive ice sheets.

Testing across various acceleration factors in the three coupled experiment classes has also revealed a trade-off between computational efficiency and integrity in melting response. While higher acceleration factors reduce simulation duration more, they also introduce larger deviations in melting response. This necessitates a careful balance between computational efficiency 580 and the integrity of the modelled melting response.

While we have used a fixed cavity residence time to interpret our experiment results, the cavity residence time in coupled models varies due to cavity geometry and circulation changes. This poses a challenge when using the accelerated forcing approach: the basal melting integrity maintained for one acceleration factor might not hold for another. Time-varying acceleration factors could address the challenge and require exploration in future developments.

Last, we emphasize that applying the accelerated forcing approach and choosing the acceleration factor should be evaluated case-by-case, with careful judgment and sensitivity testing.

*Code availability.* The coupled model used the ice sheet model Elmer/Ice Version 9.0 (https://github.com/ElmerCSC/elmerfem.git;Gagliardini et al. (2013)), the ocean model FVCOM ( https://github.com/UK-FVCOM-Usergroup/uk-fvcom/tree/akvaplan_dev, Zhou and Hattermann (2020) ), the ocean model ROMSIceShelf Version:1.0 with code (https://doi.org/10.5281/zenodo.3526801; Galton-Fenzi (2009) ), and the 590 coupled framework FISOC Version 1.1 (https://doi.org/10.5281/zenodo.4507182; Gladstone et al. (2021)). The FISOC-ROMSIceShelf-Elmer/Ice source code and input files needed to run the ROMS-based coupled model experiments in this study are all publicly available (https://doi.org/10.5281/zenodo.5908713, Zhao et al. (2022)). The FISOC-FVCOM-Elmer/Ice model shares the same ice sheet model input files as the FISOC-ROMS-Elmer/Ice model. The FVCOM-based coupled simulations use the same ice sheet model input files as the ROMS-based coupled simulations. The ocean model input files and model results for the FVCOM-based simulations are all publicly 595 preserved at the Norwegian national research data archive and can be downloaded anonymously by anyone via a web-based interface (https://doi.org/10.11582/2024.00122).

*Author contributions.* QZ, RG, and TH conceptualized and designed the study, CZ and BGZ contributed to the experiment design. QZ implemented the experiments using FVCOM, and CZ implemented the experiments using ROMS and Elmer/Ice. QZ led the analysis and wrote the initial draft. All authors contributed to the discussion of the results and paper writing.

*Competing interests.* The authors declare that they have no conflict of interest.

*Acknowledgements.* Qin Zhou received financial support from the Norwegian Research Council under projects 295075 and 343397. Chen Zhao was supported by the Australian Research Council Discovery Early Career Researcher Award (DE240100267) and the Australian

Government through the Antarctic Science Collaboration Initiative program (ASCI000002). Benjamin Galton-Fenzi also received funding from the Antarctic Science Collaboration Initiative program (ASCI000002). Tore Hattermann received financial support from the Norwegian Research Council project 280727. Rupert Gladstone was supported by the Academy of Finland grants 322430 and 355572, and by the Finnish Ministry of Education and Culture and CSC - IT Center for Science (Decision diary number OKM/10/524/2022). David Gwyther was supported by the Australian Research Council Discovery Project Grant DP220102525. The authors wish to acknowledge Sigma2 HPC, Norway, and CSC – IT Center for Science, Finland, for providing computational resources.

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
