# Peer review of "Evaluating an accelerated forcing approach for improving computational efficiency in coupled ice sheet-ocean modelling"

_Geoscientific Model Development, 2023_

## Referee Comment (RC1)

Comments on "Evaluating an accelerated forcing approach for improving computational efficiency in coupled ice sheet-ocean modelling" by Qin Zhou and colleagues.

This study proposes and evaluates a method to accelerate ocean–ice-sheet coupled simulations by considering that ocean simulations represent longer time periods than the mode time and by providing accelerated changes in ice geometry to the ocean model. Computational cost is a strong limitation of ocean–ice-sheet coupled models for sea level projections, so it is an important investigation. However, I am not convinced that "this approach could be applicable in modelling studies related to Antarctica's contribution to sea level rise projections" for the reasons below. This is a very important aspect that should be clarified before modelling groups start implementing this approach.

**Major comments**

I have two important concerns with the applicability to real world simulations, which should be discussed and probably reflected in the abstract:

1- Numerous studies have highlighted the significant impact of ice-shelf and iceberg meltwater on the ocean stratification, with important consequences for the evolution of sea ice (Bintanja et al, 2013; Swart and Fyfe 2013; Merino et al., 2018), Antarctic bottom water formation (Li et al., 2023), ocean currents around Antarctica (Moorman et al., 2020) and global climate (Bronselaer et al., 2018; Purich and England, 2023). If a global ocean model representing ice-shelf cavities is run with the accelerated approach over something like a (real) century, the total freshwater flux into the Southern Ocean won't be the same as in the regular simulation, which may significantly affect the climate system. Similarly, in some coupled ocean-ice sheet models like in Smith et al. (2021), the ice-sheet model sends its calving flux to the ocean model; how could this work with the accelerated approach? I guess that all these fluxes could be multiplied by alpha, but this would change the ocean dynamics. I am also unsure how it would work with an atmospheric forcing (which is absent from the idealised configurations presented here).

2- This work evaluates the accelerated forcing approach with two periods of variability: 0.6 year and 30 years (in real years). It is clearly shown that the accelerated method does not well capture the changes in response to the 30-year forcing (Fig.11). How about periods of 2-7 years that correspond both to ENSO (which significantly influences regions like the Amundsen Sea) and is closer to the residence time? Isn't it an important issue that this range is poorly represented by the accelerated method.

**Minor comments and edits:**

- L. 22: this is not only a carbon emission scenario, there are other anthropogenic emissions.

- L. 24: a better or complementary reference on the uncertainty is Seroussi et al. (2023).

- L. 30: "local" (instead of "regional") would be more in line with the results cited here (the increase is relatively small at the scale of an ice shelf).

- L. 40: "primarily in testing phases or for sensitivity studies (Muntjewerf et al., 2021)" is not so relevant for UKESM which has been used for scenario-based projections by Siahaan et al. (2022) even if there are important model biases. Furthermore, I don't understand the reference to Muntjewerf's paper which is about the Greenland ice sheet.

- L. 57: replace "Specifically" with something like "In this case" or "Under this assumption".

- L. 59-62: the formulations $\overline{\dot{z}_d(t)}$ and $\overline{\dot{z}_d(t/\alpha)}$ are not clear to me as the bar indicates a time average. Wouldn't $\overline{\dot{z}_d}^T$ and $\overline{\dot{z}_d}^{T/\alpha}$ be clearer?

- L. 66-84: at this stage, the reader does not know that you are using the ISOMIP+/MISOMIP1 configurations, so "boundary conditions" may refer to the surface boundary conditions (especially for a global ocean model) as well as the ocean lateral boundary conditions. Similarly, "far field" is not so clear at this stage.

- L. 99 & L. 104: these equations are not so clear to me. Why not using two variables for the model time ($t_M$) and the represented time ($t_R$).

- Table 4: I am not sure that averaging the barotropic stream function is the most accurate way to calculate the residence time because this function is defined in a relative way (only its gradients are physical). Taking the maximum minus the minimum seems more relevant. I am also wondering whether the relevant time in the ISOMIP+ case is the residence time in the entire rectangular domain.

- L. 216: correct "Notably,Although".

- L. 219: another very ood reference for this is Jenkins et al. (2018).

- Fig. 6 is interesting. Do the authors have an explanation for the weaker melt at the frequency of the barotropic circulation? On the left of the plot, the ocean temperature does not have time to adjust in the water entering the cavity ends up at a temperature of 0.5($T_C$+$T_W$). Towards the right of the plot (and beyond), the temperatures tend to follow the oscillatory forcing (equation 7 of the manuscript). If you assume a melt dependency to the quadratic thermal forcing and average the melt rate over time, you can probably explain the left-right asymmetry. My guess for the low central value is that the melt-induced circulation starts to increase in response to thermal forcing just when the forcing switches back to cold condition, which quickly cools the cavity, while the return to a warm phase is slower due to the low melt-induced circulation in cold conditions. In this case, the mean temperature in the cavity is closer to $T_C$, so melting is at its weakest value.

- Fig. 10, panel a: explain PFast1-mm in the caption.

- Fig. 10, panel b: the yellow red curves seem to show the relative difference (in %), not the absolute difference as indicated in the caption. Showing ΔV for the three experiments as in Fig. 11 (not the relative difference) would probably be easier to read. I also don't understand the values: why don't PFast3 and PFast10 start with 0% difference at month zero.

- L. 392: "Here exists a few locations" -> exist ?

- L. 401-404: I find this sentence hard to follow.

- L. 476: I do see reasons, see my main comments.

L. 455-468: Ok but the real ocean has a lot of variability associated with periods between 1 year and 30 years (e.g., El Niño Southern Oscillation; Holland et al., 2019). For this reason, Fig. 11 is quite concerning for an application to a real ocean.

The method should be compared to Lofverstrom et al. (2020) who present an approach for the atmosphere forcing of Greenland, but has some similarities with the method presented here.

**References**

Jenkins, A., Shoosmith, D., Dutrieux, P., Jacobs, S., Kim, T. W., Lee, S. H. and others (2018). West Antarctic Ice Sheet retreat in the Amundsen Sea driven by decadal oceanic variability. *Nature Geoscience*, *11*(10), 733-738.

Lofverstrom, M., Fyke, J. G., Thayer-Calder, K., Muntjewerf, L., Vizcaino, M. and others (2020). An efficient ice sheet/Earth system model spin-up procedure for CESM2-CISM2: Description, evaluation, and broader applicability. *Journal of Advances in Modeling Earth Systems*, *12*(8), e2019MS001984.

Seroussi, H., Verjans, V., Nowicki, S., Payne, A. J., Goelzer, H., Lipscomb, W. H. and others (2023). Insights into the vulnerability of Antarctic glaciers from the ISMIP6 ice sheet model ensemble and associated uncertainty, *The Cryosphere*, 17, 5197–5217.

Siahaan, A., Smith, R. S., Holland, P. R., Jenkins, A., Gregory, J. M. and others (2022). The Antarctic contribution to 21st-century sea-level rise predicted by the UK Earth System Model with an interactive ice sheet. *The Cryosphere*, *16*(10), 4053-4086.

---

## Referee Comment (RC2)

[referee-annotated manuscript omitted]

---

## Author Comment (AC1)

**Reviewer #3**

**General comments**

*The Authors seek to provide a solution to a current problem limiting the use of coupled ice-ocean models, namely the increased computational expense of the ocean part of the model when compared to the ice side. I find this a very worthwhile and relevant topic suitable for the journal. The justification, methodology and results are well presented. I feel that at present the authors are slightly over selling the potential use of their method without some further additions and clarifications within the discussion section.*

We appreciate the reviewer's positive feedback and recognition of the relevance of our study. In response to your and other reviewers' concerns regarding the potential application of the accelerated forcing approach, we have expanded substantially our discussion in the revised manuscript. This includes a more detailed examination of key limitations such as freshwater inconsistency when using the acceleration approach and the approach's challenges in capturing sub-decadal and decadal ocean forcing variability.

*1) In the current model framework, ice calving and the resultant freshwater input to the ocean is ignored. Similarly for any models that use real freshwater fluxes on the ocean time step. Do the authors envisage any potential problems with their accelerated forcing scheme if such processes were to be included?*

This was also pointed out by Reviewers #1 and #2. We have included a paragraph in the discussion section of the revised manuscript to address the inconsistencies in freshwater and energy that arise when using the accelerated forcing approach, as

*"It is important to acknowledge the limitations of our idealized study. When investigating the sensitivity of melting responses to changes in the timescale of the boundary conditions, we have only considered the lateral ocean conditions and changes in the ice draft, assuming these factors predominantly control the cavity circulation and, thus, the basal melting. This simplification presents challenges when applied to real-world scenarios where other boundary conditions affecting the cavity properties, as well as the open ocean, can not be ignored. One of them is the total glacial meltwater input to the ocean, comprising melt due to iceberg calving, basal melting, and subglacial discharge (from the subglacial hydrologic system). Numerous studies have highlighted the significant impact of glacial meltwater on ocean stratification, with important consequences for the evolution of sea ice(Bintanja et al., 2013; Merino et al., 2018; Goldberg et al., 2023), Antarctic bottom water formation (Li et al., 2023), ocean currents around Antarctica (Nakayama et al., 2021; Gwyther et al., 2023; Moorman et al., 2020), (Bronselaer et al., 2018; Purich and England, 2023; Li et al., 2024). The current study, which focuses on fine-resolution ice sheet-ocean interactions at the*

*Antarctic margins, specifically the ice shelf cavity, includes only the ocean-driven melt component of glacial meltwater. This is because basal meltwater has the largest impact on cavity circulation, mainly through buoyancy forcing. Larger-scale studies would also need to quantify the impact of other components of glacial meltwater, especially the calving flux, under accelerated forcing.* "

"*Furthermore, adjustments in glacial meltwater input are necessary to realistically represent its impacts on ocean and climate under the accelerated forcing. Without such adjustments, the total freshwater flux into the Southern Ocean would not be consistent with that under the regular forcing, potentially distorting climate simulations. However, accelerating the meltwater flux introduces its own challenges. A significant increase in local freshwater input over a short period can drastically alter local salinity gradients and stratification. This disruption can affect everything from mixing processes to ocean currents, potentially leading to unrealistic model behavior. Following Lofverstrom et al. (2020), we propose not accelerating the meltwater flux in order to maintain realistic local ocean dynamics. Instead, to mitigate the inconsistent freshwater input in the accelerated simulations, we suggest applying periodic restoration techniques to adjust the ocean's salinity and temperature fields using observed or targeted values (Griffies et al., 2009, 2016; Lofverstrom et al., 2020). Moreover, we expect similar inconsistencies in atmospheric boundary conditions—such as precipitation (freshwater input), and wind and radiation fluxes (energy input)-under the accelerated forcing. The aforementioned periodic restoration techniques can also help reduce the effects of these inconsistencies, thereby ensuring more representative freshwater and energy inputs in the ocean model.*"

*2) Likewise, the calving front is currently fixed in time. if it were to move in time would the approach still hold? My inclination would be that 1) and 2) are not deal breakers, but I would appreciate some discussion about them.*

This is a valid concern. The applicability of the accelerated forcing approach in the case of dynamic calving has not been tested in our study due to the model complexity when calving parameterisations are added. Implementation of dynamic calving in coupled ice sheet - ocean models is cutting edge, with very few research teams having stably implemented this capability (Asay-Davis et al., 2016), which may introduce additional uncertainties in evaluating the accelerated forcing approach. Our approach specifically presents ice draft change to the ocean model in the form of a rate over time, so the natural way to implement ice front movement would also be through very rapid thinning (corresponding to retreat) or thickening (advance), which brings its own challenges that need to be understood before assessing the coupled system in the context of an evolving calving front. Furthermore, given that the total calving flux is likely on the same order of magnitude as the sub ice shelf melt flux, but likely injected to the ocean over a larger area (as icebergs decay gradually during their drift), we do not anticipate that incorporating a calving flux into the coupled system would have a greater impact than the already included sub ice shelf melting. However,

we acknowledge the importance of investigating this aspect. The IceOcean2 experiment in the MISOMIP1 framework, which includes a dynamic calving front, provides an ideal framework for such an evaluation. Thus, we recommend future studies to evaluate the accelerated forcing approach when dynamic calving is activated, focusing on the response of the ocean to the additional calving freshwater input. The following discussion has been included in the discussion section of the revised manuscript:

"*Our study demonstrates that the accelerated forcing approach can directly contribute to the MISOMIP1 project by reducing the required simulation time of 100 years, depending on the acceleration factors. Applying the accelerated approach with a factor of 3 for the IceOcean1 experiment has reduced the spun-up simulation duration by a factor of 3 and reproduced most of the melting diagnostics within 10% of those with the regular forcing approach across two participating coupled models. Recommending the accelerated forcing approach to other participating models within the MISOMIP framework would provide a more comprehensive understanding of the robustness and applicability of the approach in idealized model setups. Furthermore, our current evaluation of the approach has been conducted using the IceOcean1 setup, where the calving front is fixed. It would be worthwhile to explore the applicability of the accelerated forcing approach using the IceOcean2 setup, which is similar to the IceOcean1 setup but includes dynamic calving. This extension could enhance our understanding of the ocean's response to calving fluxes under accelerated forcing, which will be discussed in detail in the following paragraph.* "

*3) In regards to ice dynamics melting near or at the grounding line is of crucial importance to represent accurately. As such it is a potential concern that the differences when using the acceleration factor are located in such areas. It would be good to see further discussion on this topic included.*

We agree with the reviewer that melting near or at the grounding line is of crucial importance to represent accurately. This is why we have chosen to also examine integrated ice draft changes and grounding line position at the end of the simulation. Specifically, we have added a plot of grounding line evolution to demonstrate that this metric is relatively robust to accelerated forcing (Figure 1). Our slowly evolving simulations based on the spun-up cavity in the Constant experiment class show identical grounding line retreat after 100 years for different acceleration factors (Figure 1a), with only minor variations in the timing of ungrounding of individual model grid elements.

We have included additional statements in the revised manuscript to justify our choice of diagnostics in the subsection on evaluating the accelerated forcing approach: " *In addition, since the absolute differences in ocean-driven melting from the accelerated forcing concentrate close to the grounding line across all experiment classes—a detail that will be elaborated on below—and considering that marine ice sheets are sensitive to melt patterns near the grounding line, we*

[Figure]

Figure 1: Time series of grounding line positions along the central domain (y = 40 km) across all simulations in (a) FVCOM-based Constant class, (b)ROMS-based Constant class, (c) Periodic-fast class, and (d) Periodic-slow class.

*also evaluate integrated ice draft changes and grounding line positions through-
out the simulation. These metrics allow us to assess the net effect of melting
differences near the grouding line on ice dynamics under accelerated forcing. "*

*4) At present domain wide volume changes are shown from the ocean side only.
It would be good to see some plots of ice Volume Above Floatation to get an idea
of the relative impacts of the acceleration method upon sea level predictions. It
would also be good to see some measure of the rate of grounding line retreat over
time. Perhaps along a central profile, or a measure of domain grounded area?*

We appreciate the reviewer's suggestion to enrich our analysis with plots of ice
Volume Above Floatation (VAF) and some measures on the grounding line re-
treat. These additions would indeed provide valuable insights into the broader
impacts of the accelerated forcing approach on sea-level predictions and ice dy-
namics.

In the revised manuscript, we have added plots of time series of grounding line
positions along the central line of the domain (y = 40 km) across all experiment
classes, as shown in Figure 1.

Unfortunately, we need to rerun the simulations to calculate the VAF. Given
that our study primarily focuses on the oceanic response under accelerated forc-
ing, not direct ice sheet behavior, we did not include VAF calculations in our
simulations. If necessary, we can rerun some of the key simulations to include
the VAF calculations. However, for the current study, we believe that the pro-
vided results sufficiently address the intended scope.

We hope that the additional grounding line measures will suffice for the aims
of this study, and we appreciate your understanding of the practical limitations
related to calculating the VAF.

*If the above points are addressed, as well as those of the other reviewers (with
who I find myself in agreement) I am in agreement with), I would be happy to
recommend publication.*

We are grateful for your support of our manuscript. We have addressed the
points you highlighted, as well as the concerns raised by the other reviewers, in
our revised submission. We hope that these revisions meet your expectations
and look forward to your final recommendations.

**Minor comments typos:**

*L 79 "circulation to flush"*

Corrected.

*L 85 I think a brief mention of the model domain to be used should be included here to help orientate the reader, as it is a little ambiguous what is meant by the MISOMIP1 framework.*

The following explanations have been added to the end of the introduction section in the revised manuscript:

"*Section 4 assesses the accelerated forcing approach across three scenarios, employing an idealized coupled model setup consistent with the MISOMIP1 project. This setup features a single, idealized ice shelf and excludes interactions with the atmosphere and sea ice (Asay-Davis et al., 2016).*"

*L 175 Is there a reason that different oscillation periods are being used for each set up?*

Yes, our selection of varied periods was intentional. We aimed to explore periods that are significantly shorter than, comparable to, and much longer than the mean cavity residence time. Ideally, incorporating a broader range of periods would provide a more comprehensive quantification of the melting response to transient ocean forcing. However, due to limitations in computational resources, we restricted our experiments to the periods presented in the manuscript.

*L 216 'Notably, although..'*

Corrected.

*L 401 'with the exception of the relative...changes exceding 10%.....'*

We have rephrased this sentence for clarity in the revised version of the manuscript, as
" In summary, our analyses show that the accelerated forcing simulations generally reproduce the time-averaged melting response, overall ocean volume changes, spatial distributions of melt rates, and integrated ice draft changes. The relative changes in these variables are kept under 10% across most locations. However, at a few locations near the grounding line, relative differences in melt rates and integrated ice draft changes exceed 10% when a higher acceleration factor of 10 is used. Thus, we consider the accelerated forcing approach to be suitable when the forcing timescale is significantly shorter than the cavity residence time, as suggested by our findings from the stand-alone experiments."

*Figure 2 Caption - 'curry colored' could be confusing for non native english speakers.*

Modified.

**References**

Asay-Davis, X.S., Cornford, S.L., Durand, G., Galton-Fenzi, B.K., Gladstone, R.M., Gudmundsson, G.H., Hattermann, T., Holland, D.M., Holland, D., Holland, P.R., Martin, D.F., Mathiot, P., Pattyn, F., Seroussi, H., 2016. Experimental design for three interrelated marine ice sheet and ocean model intercomparison projects: MISMIP v. 3 (MISMIPC), ISOMIP v. 2 (ISOMIPC) and MISOMIP v. 1 (MISOMIP1). Geosci. Model Dev. 9, 2471–2497.

Bintanja, R., van Oldenborgh, G.J., Drijfhout, S., Wouters, B., Katsman, C., 2013. Important role for ocean warming and increased ice-shelf melt in antarctic sea-ice expansion. Nature Geoscience 6, 376–379.

Bronselaer, B., Winton, M., Griffies, S.M., Hurlin, W.J., Rodgers, K.B., Sergienko, O.V., Stouffer, R.J., Russell, J.L., 2018. Change in future climate due to antarctic meltwater. Nature 564, 53–58.

Goldberg, D.N., Twelves, A.G., Holland, P.R., Wearing, M.G., 2023. The non-local impacts of antarctic subglacial runoff .

Griffies, S.M., Biastoch, A., Böning, C., Bryan, F., Danabasoglu, G., Chassignet, E.P., England, M.H., Gerdes, R., Haak, H., Hallberg, R.W., et al., 2009. Coordinated ocean-ice reference experiments (cores). Ocean modelling 26, 1–46.

Griffies, S.M., Danabasoglu, G., Durack, P.J., Adcroft, A.J., Balaji, V., Böning, C.W., Chassignet, E.P., Curchitser, E., Deshayes, J., Drange, H., et al., 2016. Omip contribution to cmip6: Experimental and diagnostic protocol for the physical component of the ocean model intercomparison project. Geoscientific Model Development , 3231.

Gwyther, D.E., Dow, C.F., Jendersie, S., Gourmelen, N., Galton-Fenzi, B.K., 2023. Subglacial freshwater drainage increases simulated basal melt of the totten ice shelf. Geophysical Research Letters 50, e2023GL103765.

Li, D., DeConto, R.M., Pollard, D., Hu, Y., 2024. Competing climate feedbacks of ice sheet freshwater discharge in a warming world. Nature Communications 15, 5178.

Li, Q., England, M.H., Hogg, A.M., Rintoul, S.R., Morrison, A.K., 2023. Abyssal ocean overturning slowdown and warming driven by antarctic meltwater. Nature 615, 841–847.

Lofverstrom, M., Fyke, J.G., Thayer-Calder, K., Muntjewerf, L., Vizcaino, M., Sacks, W.J., Lipscomb, W.H., Otto-Bliesner, B.L., Bradley, S.L., 2020. An efficient ice sheet/earth system model spin-up procedure for cesm2-cism2: Description, evaluation, and broader applicability. Journal of Advances in Modeling Earth Systems 12, e2019MS001984.

Merino, N., Jourdain, N.C., Le Sommer, J., Goosse, H., Mathiot, P., Durand, G., 2018. Impact of increasing antarctic glacial freshwater release on regional sea-ice cover in the southern ocean. Ocean Modelling 121, 76–89.

Moorman, R., Morrison, A.K., McC. Hogg, A., 2020. Thermal responses to antarctic ice shelf melt in an eddy-rich global ocean–sea ice model. Journal of Climate 33, 6599–6620.

Nakayama, Y., Cai, C., Seroussi, H., 2021. Impact of subglacial freshwater discharge on pine island ice shelf. Geophysical research letters 48, e2021GL093923.

Purich, A., England, M.H., 2023. Projected impacts of antarctic meltwater anomalies over the twenty-first century. Journal of Climate 36, 2703–2719.

---

## Author Comment (AC2)

**Reviewer #1**

*Comments on "Modeling ice shelf cavities in the unstructured-grid, Finite Volume Community Ocean Model: Implementation and effects of resolving small-scale topography" by Qin Zhou and colleagues.*

*This study proposes and evaluates a method to accelerate ocean–ice-sheet coupled simulations by considering that ocean simulations represent longer time periods than the mode time and by providing accelerated changes in ice geometry to the ocean model. Computational cost is a strong limitation of ocean–ice-sheet coupled models for sea level projections, so it is an important investigation. However, I am not convinced that "this approach could be applicable in modelling studies related to Antarctica's contribution to sea level rise projections" for the reasons below. This is a very important aspect that should be clarified before modelling groups start implementing this approach.*

We thank the reviewer for his valuable feedback and the opportunity to clarify our study. We acknowledge the reviewer's concern about the applicability of our approach to modelling studies related to Antarctica's contribution to sea level rise projections. Our primary objective is to introduce and explore this novel approach rather than to claim its definitive success or applicability in all scenarios. To better reflect the exploratory nature of our research, we have substantially revised the discussion section of our manuscript. This revision aims to more comprehensively discuss the relevance and limitations of our approach in modeling studies concerning Antarctica's contribution to sea level rise projections. This update is intended to convey that while our approach shows promise, it is still in the developmental phase and requires further validation and refinement. Below, we will address the reviewer's concerns in detail by responding to each of his comments.

**Major comments**

*I have two important concerns with the applicability to real world simulations, which should be discussed and probably reflected in the abstract:*

*__1-__ Numerous studies have highlighted the significant impact of ice-shelf and iceberg meltwater on the ocean stratification, with important consequences for the evolution of sea ice (Bintanja et al, 2013; Swart and Fyfe 2013; Merino et al., 2018), Antarctic bottom water formation (Li et al., 2023), ocean currents around Antarctica (Moorman et al., 2020) and global climate (Bronselaer et al., 2018; Purich and England, 2023). If a global ocean model representing ice-shelf cavities is run with the accelerated approach over something like a (real) century, the total freshwater flux into the Southern Ocean won't be the same as in the regular simulation, which may significantly affect the climate system. Similarly, in some coupled ocean-ice sheet models like in Smith et al. (2021), the ice-sheet model sends its calving flux to the ocean model; how could this work with the accelerated approach? I guess that all these fluxes could be multiplied by alpha,*

*but this would change the ocean dynamics. I am also unsure how it would work with an atmospheric forcing (which is absent from the idealised configurations presented here).*

We thank the reviewer for highlighting the critical aspect of meltwater flux when applying the accelerated forcing approach, which we did not address in our original manuscript. We agree that glacial meltwater (basal melting, calving flux, and subglacial discharge) significantly impacts many aspects of the ocean and global climate, as pointed out by the reviewer. We also agree these processes should be adequately represented when applying the accelerated forcing approach in real-world scenarios.

In our idealized simulations, we have only considered the ice draft change and far-field ocean conditions when investigating the sensitivity of basal melting response to the changes in the timescale of the boundary conditions, by assuming that these two factors predominantly control the cavity circulation and thus the basal melting. While this simplified approach has strengths, it presents challenges when applied to real-world scenarios where other influencing boundary conditions affecting the ocean are not considered, such as the glacial meltwater flux, wind, and radiation fluxes at the ocean-atmospheric interface.

Here, we take the glacial meltwater as an example, and the same applies to the precipitation/evaporation. Although accelerating the meltwater by multiplying it with the acceleration factor ensures the consistency of total freshwater input under the accelerated forcing, intense local freshwater input in a short period can disrupt local salinity gradients and stratification. This disruption can affect everything from mixing processes to ocean currents, potentially leading to unrealistic model behavior. Conversely, not accelerating the meltwater maintains realistic stratification for local processes but doesn't conserve total freshwater input, leading to inconsistencies over the long term.

To address this, we propose not accelerating the meltwater flux to maintain realistic local ocean dynamics. In stead, we suggest applying periodic restoration techniques to adjust the ocean's salinity and temperature field using observed or targeted values to mitigate the inconsistent freshwater input in the accelerated simulations. A similar technique has successfully been used in asynchronous coupling between ice sheets and climate models to reduce artificial drift in the ocean caused by inconsistent global freshwater input Lofverstrom et al. (2020).

For other atmospheric conditions, such as wind stress and heat fluxes, we also propose not accelerating the absolute values as it would lead to unrealistic and non-physical results. The same periodic restoration techniques can be used to mitigate the inconsistency of freshwater and energy input to the ocean due to not-accelerated atmospheric boundary conditions under the accelerating forcing. However, the full exploration of the impacts of these inconsistencies on the ocean and climate system extends beyond the scope of this study. Nevertheless,

we have discussed these trade-offs and potential solutions in the discussion section of our revised manuscript, as

"*It is important to acknowledge the limitations of our idealized study. When investigating the sensitivity of melting responses to changes in the timescale of the boundary conditions, we have only considered the lateral ocean conditions and changes in the ice draft, assuming these factors predominantly control the cavity circulation and, thus, the basal melting. This simplification presents challenges when applied to real-world scenarios where other boundary conditions affecting the cavity properties, as well as the open ocean, can not be ignored. One of them is the total glacial meltwater input to the ocean, comprising melt due to iceberg calving, basal melting, and subglacial discharge (from the subglacial hydrologic system). Numerous studies have highlighted the significant impact of glacial meltwater on ocean stratification, with important consequences for the evolution of sea ice(Bintanja et al., 2013; Merino et al., 2018; Goldberg et al., 2023), Antarctic bottom water formation (Li et al., 2023), ocean currents around Antarctica (Nakayama et al., 2021; Gwyther et al., 2023; Moorman et al., 2020), (Bronselaer et al., 2018; Purich and England, 2023; Li et al., 2024). The current study, which focuses on fine-resolution ice sheet-ocean interactions at the Antarctic margins, specifically the ice shelf cavity, includes only the ocean-driven melt component of glacial meltwater. This is because basal meltwater has the largest impact on cavity circulation, mainly through buoyancy forcing. Larger-scale studies would also need to quantify the impact of other components of glacial meltwater, especially the calving flux, under accelerated forcing.*"

"*Furthermore, adjustments in glacial meltwater input are necessary to realistically represent its impacts on ocean and climate under the accelerated forcing. Without such adjustments, the total freshwater flux into the Southern Ocean would not be consistent with that under the regular forcing, potentially distorting climate simulations. However, accelerating the meltwater flux introduces its own challenges. A significant increase in local freshwater input over a short period can drastically alter local salinity gradients and stratification. This disruption can affect everything from mixing processes to ocean currents, potentially leading to unrealistic model behavior. Following Lofverstrom et al. (2020), we propose not accelerating the meltwater flux in order to maintain realistic local ocean dynamics. Instead, to mitigate the inconsistent freshwater input in the accelerated simulations, we suggest applying periodic restoration techniques to adjust the ocean's salinity and temperature fields using observed or targeted values (Griffies et al., 2009, 2016; Lofverstrom et al., 2020). Moreover, we expect similar inconsistencies in atmospheric boundary conditions—such as precipitation (freshwater input), and wind and radiation fluxes (energy input)-under the accelerated forcing. The aforementioned periodic restoration techniques can also help reduce the effects of these inconsistencies, thereby ensuring more representative freshwater and energy inputs in the ocean model.*"

**2-** *This work evaluates the accelerated forcing approach with two periods of vari-*

*ability: 0.6 years and 30 years (in real years). It is clearly shown that the accelerated method does not well capture the changes in response to the 30-year forcing (Fig.11). How about periods of 2-7 years that correspond both to ENSO (which significantly influences regions like the Amundsen Sea) and is closer to the residence time? Isn't it an important issue that this range is poorly represented by the accelerated method.*

We agree with the reviewer that another limitation of the accelerated approach lies in its poor representation of melting response to oceanic forcing of periodicity of sub-decades and decades in the accelerated forcing simulations because this range might be either close to the cavity residence time of the cold-water ice shelves or any acceleration of the timescale would be close to the mean cavity residence times. Given that forcing variability of these timescales significantly influences regions like the Amundsen Sea (Jenkins et al., 2018; Huguenin et al., 2024), we have added the discussion of this limitation in the revised version of the manuscript, as

*The mean cavity residence time, mainly determined by the cavity geometry and barotropic transport, is an intrinsic timescale of the ocean model. It represents the time needed for the cavity to reach an equilibrium melting state, where the cavity is filled with water that is exactly in balance with the steady ocean forcing (Holland, 2017). When the timescale of unsteady ocean forcing approaches this intrinsic timescale, interactions occur between basal melting, cavity circulation, heat inertia within the cavity, and transient changes in boundary forcing. Consequently, the melting response becomes highly sensitive to any alterations in these factors. This scenario challenges the underlying assumption of the accelerated forcing approach that basal melting response is not sensitive to corresponding accelerations in ocean boundary forcing. Hence, the accelerated forcing approach loses applicability when the forcing timescale, whether under regular or accelerated forcing, is in the order of the mean cavity residence time. This finding limits the approach's applications in real-world scenarios. For example, the El Niño-Southern Oscillation (ENSO), which significantly influences regions like the Amundsen Sea (Paolo et al., 2018; Huguenin et al., 2024), may be poorly represented under accelerated forcing due to its typical 2-7 year cycle coinciding with the cavity residence time of certain ice shelves around Antarctica. Notably, cold-water shelves like the Fichner-Ronne Ice Shelf and the Ross Ice Shelf have a cavity residence time of 4-8 years (Nicholls and Østerhus, 2004; Loose et al., 2009), and warm-water shelves like those in the Amundsen Sea have even shorter cavity residence times given their smaller sizes and faster melting-driven cavity circulations. This alignment could lead to an overestimation of the melting response when ENSO's timescale is compressed under accelerated forcing. Moreover, even if the multi-decadal variation in forcing substantially exceeds the cavity residence time, applying the accelerated forcing approach may result in an underestimation of basal melting response once its compressed timescale is comparable to the cavity residence time. Therefore, caution should be used when applying the accelerated forcing approach to studies addressing climate variabil-*

*ity on sub-decadal to decadal timescales. "*

**Specific Comments**

*-L. 22: this is not only a carbon emission scenario, there are other anthropogenic emissions.*

We have removed 'carbon' in the sentence to broaden the reference to emissions to include not just carbon but also other anthropogenic emissions that contribute to climate change.

*-L. 24: a better or complementary reference on the uncertainty is Seroussi et al. (2023).*

The reference has been added.

*-L.30: "local" (instead of "regional") would be more in line with the results cited here (the increase is relatively small at the scale of an ice shelf).*

We have replaced "regional" with "local" in the sentence.

*-L. 40: "primarily in testing phases or for sensitivity studies (Muntjewerf et al., 2021)" is not so relevant for UKESM which has been used for scenario-based projections by Siahaan et al.(2022) even if there are important model biases. Furthermore, I don't understand the reference to Muntjewerf's paper which is about the Greenland ice sheet.*

We agree with the reviewer that "primarily in testing phases or for sensitivity studies (Muntjewerf et al., 2021)" is not so relevant in this context. We have removed it and instead cited Sianhaan et al's work to support the preceding statement. The revised statement now reads: " *More recently, coupled ice sheet-ocean model configurations on the circumpolar scale or beyond, with cavities explicitly resolved, have begun to emerge (Smith et al., 2021; Pelletier et al., 2022; Siahaan et al., 2022).*"

*-L.57: replace "Specifically" with something like "In this case" or "Under this assumption".*

We have replaced "Specifically" with "Under this assumption".

*-L. 59-62: the formulations $\overline{\dot{z}_d(t)}$ and $\overline{\dot{z}_d(t/\alpha)}$ are not clear to me as the bar indicates a time average. Would not $\overline{\dot{z}_d}^T$ and $\overline{\dot{z}_d}^{T/\alpha}$ be clearer?*

We have incorporated the reviewer's suggestion to add subscripts indicating time averages. Furthermore, we have revised the notation for the oceanic effect

on ice draft change to avoid confusion with the total ice draft change when introducing the data flow within the coupled system later in the text, as

*" Under this assumption, within the total ice draft change $\Delta z_d$, which includes contributions from ocean-driven change and ice-dynamics-driven change $\Delta z_{d_i}$, the ocean-driven draft change can be expressed as an integral of basal melt rate $M$ over the coupling time interval $T$, as*

$$\Delta z_d = \int^T M dt + \Delta z_{d_i}. \tag{1}$$

*The ocean-driven change can be further expressed as the time integral of a quasi-steady-state mean melt rate $\overline{M}^T$ over the coupling interval $T$, as*

$$\int^T M dt = \overline{M}^T \cdot T. \tag{2}$$

*By assuming that the mean melt rate $\overline{M}^T$ during the coupling interval $T$ can be approximated by a quasi-steady-state melt rate $\overline{M}^{T/\alpha}$ during a shortened coupling interval of $T/\alpha$, the ocean model simulation duration can be reduced from $T$ to $T/\alpha$, hereby accelerating the timescale of the ocean model by a factor of $\alpha$. Note that the superscripts $T$ and $T/\alpha$ denote the coupling intervals, not the exponents or powers of a number. In addition, to maintain the model's integrity under the accelerated approach, the timescales of the ocean model's boundary conditions should be also accelerated accordingly to accommodate the timescale change from $T$ to $T/\alpha$."*

*-L.66-84: at this stage, the reader does not know that you are using the ISOMIP+/MISOMIP1 configurations, so "boundary conditions" may refer to the surface boundary conditions (especially for a global ocean model) as well as the ocean lateral boundary conditions. Similarly, "far field" is not so clear at this stage.*

We appreciate the reviewer's observation regarding the potential ambiguity of 'boundary conditions' and 'far field' at this point in the manuscript. In response, we have revised our text, to begin with a general introduction to the various boundary conditions a coupled ice sheet-ocean model system is subject to, then specifically narrow down to the two boundary conditions central to our investigation. The revised text now reads:

*"In a coupled ice sheet-ocean model system, the ocean model is subject to a range of boundary conditions: changes in ice draft and meltwater flux at the ice sheet-ocean interface, momentum, freshwater, and radiation fluxes at the atmosphere-ocean interface, and lateral ocean conditions. In this study, we only focus on the lateral ocean conditions and the ice draft change at the ice sheet-ocean interface, as these two factors predominantly control the cavity circulation*

*and, thus, the basal melting response."*

In addition, we have also moved the term of "far field" at this point to avoid confusion.

*-L. 99 & L. 104: these equations are not so clear to me. Why not using two variables for the model time ($t_M$) and the represented time ($t_R$).*

We appreciate the reviewer's suggestion to use distinct variables for the model time and the represented time. However, we have opted to maintain our current notation of $t$ and $t/\alpha$ for a couple of reasons. First, using $t$ and $t/\alpha$ conveys the concept of compressed time, which is central to understanding the accelerated forcing approach. Secondly, introducing additional variables could potentially complicate the notation without adding significant clarity. We aim to keep the explanation as straightforward as possible while adequately conveying the necessary concepts.

Equations 3 and 4 are consistent with those used in Gladstone et al. (2021) because we employ the same coupling framework in our study, ensuring alignment and comparability of methodologies.

*-Table 4: I am not sure that averaging the barotropic stream function is the most accurate way to calculate the residence time because this function is defined in a relative way (only its gradients are physical). Taking the maximum minus the minimum seems more relevant. I am also wondering whether the relevant time in the ISOMIP+ case is the residence time in the entire rectangular domain.*

We appreciate the reviewer's concern regarding the method we employed to calculate the cavity residence time by averaging the barotropic stream function. Our choice to use this method was guided by its application in the study by Holland (2017), which inspired the design of our experiments. While we acknowledge that other methods might offer different insights into cavity dynamics, for the purposes of our current study, we believe this approach serves our study's objectives, providing a reliable measure of cavity residence time.

*-L. 216: correct "Notably,Although".*

Corrected.

*-L. 219: another very good reference for this is Jenkins et al. (2018).*

The reference is added.

*-Fig. 6 is interesting. Do the authors have an explanation for the weaker melt at the frequency of the barotropic circulation? On the left of the plot, the ocean temperature does not have time to adjust in the water entering the cavity ends*

*up at a temperature of 0.5(TC+TW). Towards the right of the plot (and beyond), the temperatures tend to follow the oscillatory forcing (equation 7 of the manuscript). If you assume a melt dependency to the quadratic thermal forcing and average the melt rate over time, you can probably explain the left-right asymmetry. My guess for the low central value is that the melt-induced circulation starts to increase in response to thermal forcing just when the forcing switches back to cold condition, which quickly cools the cavity, while the return to a warm phase is slower due to the low melt-induced circulation in cold conditions. In this case, the mean temperature in the cavity is closer to TC, so melting is at its weakest value.*

We thank the reviewer for the insightful interpretation of Fig.6. Your comments help deepen our understanding of the observed phenomena in the plot. In our manuscript, we discussed why the melting response tends to stabilize on the left side of the plot, explaining that " This is because multiple COLD and WARM waters coexist within the cavity in this regime, effectively canceling each other in the spatial mean, leading to a melting response close to that from the MEAN forcing simulation. ". However, We have not explained the weaker melt at the frequency of the barotropic circulation or the left-right asymmetry.

We have now enhanced our explanation of Fig.6 in the revised version of the manuscript by incorporating your interpretation, as

*"Figure 6 not only reinforces the three distinct melting regimes observed in the time series and spatial distribution figures but also provides additional insights for predetermining suitable scenarios for the accelerated forcing approach. First, the normalized melt rates reach their minimum across all three model configurations when the oscillation periods approximate the MCRTs (Log2(Normalized timescale) $\simeq$ 0). In this regime, melt-induced circulation begins to increase with the warm phase of the oscillation just as the forcing shifts back to the cold phase, which rapidly cools the cavity. The return to the warm phase is slower due to diminished melt-induced circulation in cold conditions, resulting in a cavity temperature closer to the COLD forcing, thereby minimizing melting. This suggests that when the oscillation period of ocean forcing, either accelerated or not, approximates the MCRT, the melting response is likely to deviate significantly from the actual response, thus challenging the underlying assumption of the accelerated forcing approach. Secondly, when oscillation periods are shorter than the MCRT (Log2(Normalized timescale) ¡ 0), the melting rates tend to stabilize, as indicated by normalized melt rates clustering between 0.9 and 1.1. In this regime where the ocean conditions oscillate rapidly, the ocean temperature doesn't have time to adjust to that of the WARM or COLD profiles. This results in the water entering the cavity at a temperature close to that of the MEAN profiles, thereby leading to a melting response that is nearly equivalent to that observed under the MEAN forcing. Consequently, this response exhibits low sensitivity to rapidly varying ocean forcing. Given our earlier assertion that the accelerated forcing approach only remains valid when the basal melting response is*

*not sensitive to corresponding accelerations in ocean boundary forcing, we de-
duce the approach is applicable in this regime. In contrast, when the oscillating
forcing periods greatly exceed the MCRTs, melt rates increase significantly. In
specific, the normalized melt rates increase from about 0.7 when the forcing pe-
riod near the MCRT (Log2(Normalized timescale) $\simeq$ 0) to more than 1.1 when
the period much longer than the MCRT (Log2(Normalized timescale) ¿ 2 ) for
both ISOMIP+ domain configurations. In the FVCOM-ISOMIP+ configura-
tion, the normalized melt rate further increases to about 1.3 when the forcing
period (30 years) is seven times longer than the MCRT of 4 years. In addition,
the FVCOM-Wedge simulations display a comparable increasing trend but at a
slower rate, likely due to differences in cavity geometry. The increase in melt
rates is attributed to the quadratic relationship between melt rates and ocean
temperatures (Holland et al., 2008; Jenkins et al., 2018). In detail, as the ocean
forcing oscillates slowly, ocean temperatures tend to follow the oscillatory forcing
at every stage. When averaged over the oscillation period, the mean melt rate
aligns more closely with that from the WARM forcing and thus is higher than
that from the MEAN forcing. We expect that the melting response will stabilize
when ocean temperatures fully adjust to the oscillatory forcing. However, due
to the lack of simulations with longer periods, we are unable to determine the
minimum period necessary for the melting response to reach equilibrium at every
phase of the cycle. In scenarios where the forcing period exceeds the MCRT but
does not allow a full equilibrium melting response, the accelerated forcing ap-
proach is likely not suitable, as it tends to underestimate the melting response.*"

*- Fig. 10, panel a: explain PFast1-mm in the caption.*

We will explain it when updating the figure in the revised manuscript.

*- Fig. 10, panel b: the yellow red curves seem to show the relative difference
(in %), not the absolute difference as indicated in the caption. Showing $\delta V$ for
the three experiments as in Fig. 11 (not the relative difference) would probably
be easier to read. I also don't understand the values: why donnot PFast3 and
PFast10 start with 0 % difference at month zero.*

We agree with the reviewer that presenting $\delta V$ for the three simulations would
likely enhance readability, and we will include these changes in the revised
manuscript. Additionally, this update will avoid the issue of the unexpected
non-zero values of 0% difference at month zero. These arise because $\delta V$, while
close to zero in all three simulations, is not exactly zero. Small deviations
among the simulations can therefore lead to significant relative differences when
expressed in percentages.

*- L. 392: "Here exists a few locations" , exist ?*

Corrected.

- L. 401-404: I find this sentence hard to follow.

. We have rephrased this sentence in the revised version of the manuscript, as

" *In summary, our analyses show that the accelerated forcing simulations generally reproduce the time-averaged melting response, overall ocean volume changes, spatial distributions of melt rates, and integrated ice draft changes. The relative changes in these variables are kept under 10% across most locations. However, at a few locations near the grounding line, relative differences in melt rates and integrated ice draft changes exceed 10% when a higher acceleration factor of 10 is used. Thus, we consider the accelerated forcing approach to be suitable when the forcing timescale is significantly shorter than the cavity residence time, as suggested by our findings from the stand-alone experiments.*"

- L. 476: I do see reasons, see my main comments.

We have removed this over-selling sentence and responded to your main comments in this reply.

- L. 455-468: Ok but the real ocean has a lot of variability associated with periods between 1 year and 30 years (e.g., El Niño Southern Oscillation; Holland et al., 2019). For this reason, Fig. 11 is quite concerning for an application to a real ocean.

Your concern is valid. We have revised the paragraph substantially in the discussion section of the revised manuscript, also in response to the comments from the second reviewer, as

" *When the timescale of the ocean forcing significantly exceeds the cavity residence time, the cavity is flushed several times during each cycle. Unlike with steady ocean forcing, the cavity can never fully achieve the equilibrium melting state under oscillating ocean forcing (Holland, 2017). Nevertheless, if the period is sufficiently long, waters at each phase of the forcing cycle may have enough time to be flushed into the cavity, allowing the melting to reach a quasi-equilibrium state. This state closely approximates equilibrium but includes slight fluctuations due to the continuous variation in forcing. For instance, a period of 30 years seems long enough for the FVCOM-ISOMIP+ configuration to this quasi-equilibrium melting state at each phase of the cycle. Figure ??d illustrates that the minimum mean melt rate in FI_30yr deviates slightly from that under the COLD forcing, indicating that even the coldest waters have enough time to fill the cavity and influence melting. This suggests that warmer water phases, especially the warmest, are also sufficiently flushed into the cavity to reach a quasi-equilibrium melting state, as evidenced by the maximum mean melt rate being nearly the same as that under the WARM forcing. Considering a hypothetical 300-year forcing period, waters in each phase of the cycle would have 10 times longer to influence the cavity compared to the 30-year cycle, al-*

*lowing the quasi-equilibrium melting state in each phase to last about 10 times longer. Therefore, the melting response in any single phase of the 300-year cycle can be approximated by the response in the corresponding single phase of the 30-year cycle, supporting the fundamental assumptions of the accelerated forcing approach. While we have not tested forcings with periods longer than 30 years due to resource constraints, the constant forcing in the Constant experiment class essentially represents an infinitely slow varying force once the model reaches a quasi-steady state. This highlights the potential applicability of the accelerated forcing approach in century-long cavity-processes-oriented modelling studies, which could improve the accuracy of projections of Antarctica's contribution to sea level rise. In such projections, the slowly varying background forcing would not be periodic but instead steadily increasing at comparably slow rates in global warming scenarios. However, the linearly increasing trend from cold to warm can be considered as a warming phase of varying forcing over even longer timescales far exceeding the mean cavity residence time of any ice shelf, ensuring the applicability of the accelerated forcing approach."*

*The method should be compared to Lofverstrom et al. (2020) who present an approach for the atmosphere forcing of Greenland, but has some similarities with the method presented here.*

We thank the reviewer for pointing out this important paper. We have now added a paragraph in the introduction of the manuscript that compares our acceleration approach with previous techniques used in climate models to bridge timescale discrepancies between various model components. This includes the technique from (Lofverstrom et al., 2020), as

[revised manuscript text omitted]

---

## Author Comment (AC3)

**Reviewer #2**

**Summary**

*The study evaluates the applicability of an accelerated forcing methodology in the scope of high resolution ice-ocean coupling. The authors motivate their investigation by the typical time scale discrepancy between ice and ocean dynamics and the corresponding disparity in simulation time. First, the idea and methodology of an accelerated coupling approach is presented and the models used in the study are described.*

*Then, standalone setups of two ocean models are used to derive ice-shelf melt rates for warm, cold and mean far-field ocean forcing conditions, as well as oscillating profiles between the warm and cold case at different frequencies. The authors use the mean cavity response time (MCRT; for the mean forcing case) as a characteristic variable to evaluate the derived melt rates. They find that averaged cavity melt rates for oscillating far-field forcings that have significantly higher frequencies than the MCRT (periods ¡=10% of MCRT) are mostly in the range of 90-110% of melt rates from time averaged forcing. However, forcing frequencies that are in the same order of the MCRT or substantially lower do either under- or overestimate the mean melt rates. Based on these ocean standalone simulations the authors anticipate that the accelerating forcing approach would only work in the first case, which they subsequently test in coupled ice-ocean simulations.*

*The test scenarios for coupled ice-ocean simulations are structured in three categories: 1. constant cold-to-warm forcing as well as two periodic forcings with fast (2.) and slow (3.) varying time scales. For all categories different acceleration factors are tested (between 1.5 and 10) and evaluated to the baseline scenario (no acceleration) in terms of cavity averaged melt rates and total ocean volume changes (time series) as well as spatially fields of melt rates and ice draft changes. The authors find that acceleration works well for spun-up simulations in the constant forcing as well as the fast-varying ocean forcing case, but not in the slow-varying case.*

*Finally, the authors conclude, that their presented approach of accelerated forcing in high-resolution ocean-ice coupled models is also applicable in real-world applications for ocean to ice forcing that varies over century-long timescales.*

We thank the reviewer for the detailed and accurate summary of our study.

**General comments**

*The paper addresses a relevant and scientifically interesting topic which fits very well in the scope of GMD. It presents a novel investigation of testing the impact of asynchronous coupling in idealized setups and introduces a useful metric*

*(MCRT) to assess the applicability of the approach. The study has a sound methodology and follows a clear experimental design with valid, clear and justified assumptions. The manuscript is well written, with fluent and precise language. The title is appropriate and reflects the contents of the paper well. The abstract provides a concise and complete summary of the presented work.*

We thank the reviewer for this positive assessment.

*I have no major concerns, but a few remarks. More specific comments and suggestions for improvement are given in the attached pdf.*

We thank the reviewer for his valuable and constructive comments on improving our manuscript. we agree with the overall comments and have addressed them in detail accordingly both in the response below and in the attached PDF.

*The introduction motivates the following work well. However, more background on previous work about asynchronous ice-ocean coupling would be great to give the reader more context to the study like: What studies have used asynchronous coupling so far? Is there already some literature that compares synchronous vs asynchronous coupling?*

We appreciate the reviewer pointing out the relevance of asynchronous coupling to our study. This technique is indeed useful for bridging timescale discrepancies between ice sheets and other model components in coupled ice sheet/climate models. In response to your suggestion, we have now expanded the introduction to include a discussion of previous work on asynchronous coupling in the revised manuscript, as

" *A number of different climate related disciplines utilising coupled modelling have encountered these issues of optimising performance of a model system where individual components have varying response timescales, including atmosphere - ocean modelling (Sausen and Voss, 1996; Voss et al., 1998) and Paleoclimate modelling incorporating ice sheets (Roberts et al., 2014; Lofverstrom et al., 2020). Approaches have included "periodic synchronous coupling", where the outputs of the faster component are averaged over a short period of synchronous coupling and are then used to force the slower component(s) over a longer uncoupled period, and "asynchronous coupling", where the faster model is run for a shorter period during each coupling interval. In this context "synchronous coupling" simply means that the elapsed modelled time, measured at the time of any exchange of coupled variables, is the same for each component. This is a broader definition that has been recently used in the ice sheet - ocean community (Goldberg et al., 2018; Gladstone et al., 2021), where "synchronous coupling" has been taken to mean that both fast and slow components update the coupling variables every fast timestep. Coupling synchronicity is especially important in the regional marine ice sheet - ocean modelling community where ice shelf cavity circulation is fully resolved by the ocean model but where the*

*coupling region itself (the underside of the ice shelf) evolves with time.*"

*I have a few general remarks for plots (more specific ones are given as annotations to the pdf):*

- *for all spatial plots of ISOMIP+ domain (barotropic stream function, melt rates, ice draft changes): it would be helpful to either mark the grounding line as a line or to shade the grounded areas (e.g. light gray) to be able to distinguish regions with values close to zero and grounded areas.*

- *Do 2d plots show values out of the colorbar range (e.g. Fig 3c)? If so, please indicate this by adding out-of-range extensions to the colorbars and give maximum values in text caption.*

- *If contour spacing can't be easily inferred from colorbar, please provide this information in the caption.*

- *Please make sure that all colorbars have meaningful ticks. Fig 3b & 10c have no tick for lower bound.*

We thank the reviewer for the detailed suggestions on improving the plots. We will update all the figures in the revised manuscript according to these recommendations, as well as those provided by other reviewers.

*The simulation times (model & represented time) are all given in months. Personally I would find it more intuitive to speak of years, and where the precise number of months is important (e.g. start of spun-up simulations, etc), this information could be given in brackets in months. This remark also applies to time axis of the plots.*

We will replace 'months' with 'years' when referring to the simulation time, both in the text and the plots of the revised manuscript.

*I disagree with some statements that are made in the discussion and conclusion section:*

*The study shows that the accelerated forcing approach is not suitable when the time scale of periodical forcing is in the order of the MCRT. When the forcing period is significantly longer, mean melt rates are higher than in the mean-state (Fig. 6). I am wondering how Fig. 6 would look like if it would be extended to the right, with longer forcing periods. Would it stay constant at comparable levels like the longest tested time scales, or will it converge asymptotically to a higher value? How long would that tail be, and how big the differences? The*

*authors argue in the discussion that for a 30 year forcing period the cavity is assumed to be in equilibrium with the forcing at all times (can this be proven somehow?). But already the 20 year period deviates significantly, which is already a factor 5 higher than the MCRT of 4 years. So then the question arises, what is a minimum factor that is required to still yield realistic results? I understand that it is challenging to test much greater forcing periods due to long computation times and that this might not be feasible. However, I feel that there was little evidence given that 30 years is in equilibrium with the forcing and therefore would be same as 300 years, whereas in the same time it is shown that 20 years is already too short.*

[Figure]

Figure 1: Time series of cavity-averaged melt rates from the selected FVCOM-ISOMIP+simulations. The dashed lines extend from their respective quasi-steady state values for interpretative purposes.

We appreciate the reviewer's concerns in the discussion section (lines 456-468). We will address these concerns/questions in detail below.

- **How would Figure 6 look like when extending the x-axis further to the right?** Time series of melt rates from the 30-year forcing period, as shown in Figure 1 and also Figure 4d in the manuscript, suggest that waters at each phase of the forcing cycle may have enough time to be flushed into the cavity, allowing the melting to reach a quasi-equilibrium state. Thus, we expect that the mean melting responses from longerperiod forcing simulations would likely stay constant at comparable levels to that for the 30-year period, as it has more time to allow the waters at every phase to flush into the cavity. We have now explained this point explicitly in the discussion of the revised manuscript, as

*"When the timescale of the ocean forcing significantly exceeds the cavity residence time, the cavity is flushed several times during each cycle. Unlike with steady ocean forcing, the cavity can never fully achieve the equilibrium melting state under oscillating ocean forcing (Holland, 2017). Nevertheless, if the period is sufficiently long, waters at each phase of the forcing cycle may have enough time to be flushed into the cavity, allowing the melting to reach a quasi-equilibrium state. This state closely approximates equilibrium but includes slight fluctuations due to the continuous variation in forcing. For instance, a period of 30 years seems long enough for the FVCOM-ISOMIP+ configuration to this quasi-equilibrium melting state at each phase of the cycle. Figure 4d illustrates that the minimum mean melt rate in FI_30yr deviates slightly from that under the COLD forcing, indicating that even the coldest waters have enough time to fill the cavity and influence melting. This suggests that warmer water phases, especially the warmest, are also sufficiently flushed into the cavity to reach a quasi-equilibrium melting state, as evidenced by the maximum mean melt rate being nearly the same as that under the WARM forcing. Considering a hypothetical 300-year forcing period, waters in each phase of the cycle would have 10 times longer to influence the cavity compared to the 30-year cycle, allowing the quasi-equilibrium melting state in each phase to last about 10 times longer. Therefore, the melting response in any single phase of the 300-year cycle can be approximated by the response in the corresponding single phase of the 30-year cycle, supporting the fundamental assumptions of the accelerated forcing approach. While we have not tested forcings with periods longer than 30 years due to resource constraints, the constant forcing in the Constant experiment class essentially represents an infinitely slow varying force once the model reaches a quasi-steady state. This highlights the potential applicability of the accelerated forcing approach in century-long cavity-processes-oriented modelling studies, which could improve the accuracy of projections of Antarctica's contribution to sea level rise. In such projections, the slowly varying background forcing would not be periodic but instead steadily increasing at comparably slow rates in global warming scenarios. However, the linearly increasing trend from cold to warm can be considered as a warming phase of varying forcing over even longer timescales far exceeding the mean cavity residence time of any ice shelf, ensuring the applicability of the accelerated forcing approach."*

• Figure 1 shows that the maximum melt rates from the 20-year period forcing (FI_20yr) slightly deviate from those observed under the steady

WARM forcing (FI_C2W), unlike the 30-year period forcing (FI_30yr). This deviation suggests that even the warmest waters do not have sufficient time to fully circulate within the cavity during the 20-year cycle, and the colder waters have even less time to do so. Consequently, the normalized melt rate for the 20-year period is significantly different from that of the 30-year cycle. Our findings indicate that while the melting process can reach a quasi-equilibrium state during each phase of the 30-year cycle, it fails to do so within the 20-year cycle. We infer that the minimum period required to achieve realistic results would likely fall between 20 and 30 years.

*That also relates to my second point in the discussion/conclusion: about the application for real-world scenarios. In global warming projections/scenarios the slowly-varying background forcing would not be periodically, but rather steadily increasing at comparable slow rates. When using this acceleration method for coupled simulations, the warming rate would be increased in the accelerated ocean compared to the unaccelerated case. Also in this case it is of great importance to know what would be still acceptable rates of changing forcing without impacting the results too much. Again, here it would help if testing of more than 30 years periodic forcing is possible, as a maximum change rate can be inferred from periodic forcing. However, as this seems not feasible for the given setup/resources, linear increasing forcing from a cold to a warm state at different rates could be an option? Especially for the ice-ocean coupled setup, this would be interesting.*

We agree with the reviewer that the slowly varying background forcing would not be periodic, but rather steadily increasing at comparable slow rates in global warming scenarios. However, the linearly increasing trend from cold to warm at different rates can be considered analogous to the warming phase of oscillating forcing with different timescales. This analogy allows us to apply the conclusions drawn from periodic forcing simulations in our study to scenarios involving linearly varying forcing. The applicability of the accelerated forcing approach in these scenarios depends on the ratio of the timescale of the varying forcing to the mean cavity residence time.

To clarify, we have explicitly addressed this comment in the discussion section of the revised manuscript, as presented in our response to your previous comment above.

*Also important for the applicability of real world scenarios are the time scales of natural variability of ocean to ice forcing, e.g. at decadal timescales (Jenkins et al., 2018). I interpret the current result of the study that the MCRT for Filchner-Ronne/Ross (4-8 years) would conflict with oscillating forcing on decadal time scales. This could possibly impose major challenges for the applicability of the accelerated approach on real world scenarios. I am not certain whether this is a deal-breaker for later applications, but I would like to see much*

*more discussion about this. Stating that the ocean forcing in real world scenarios varies mostly over century-long time scales, seems a bit too simple in my view. Concerning this matter, also a discussion of mixed time scale forcing seems to be interesting, like seasonal forcing overlayed by decadal oscillations with a steady increase in the background signal.*

We agree with the reviewer that some important natural variability of the ocean forcing, such as those on sub-decadal timescales such as ENSO and decadal timescales, are expected to be poorly represented with the applied forcing approach. In response to this comment as well as the comment from the first reviewer, we have now addressed the limitation in the discussion section of the revised manuscript, as

"*The mean cavity residence time, mainly determined by the cavity geometry and barotropic transport, is an intrinsic timescale of the ocean model. It represents the time needed for the cavity to reach an equilibrium melting state, where the cavity is filled with water that is exactly in balance with the steady ocean forcing (Holland, 2017). When the timescale of unsteady ocean forcing approaches this intrinsic timescale, interactions occur between basal melting, cavity circulation, heat inertia within the cavity, and transient changes in boundary forcing. Consequently, the melting response becomes highly sensitive to any alterations in these factors. This scenario challenges the underlying assumption of the accelerated forcing approach that basal melting response is not sensitive to corresponding accelerations in ocean boundary forcing. Hence, the accelerated forcing approach loses applicability when the forcing timescale, whether under regular or accelerated forcing, is in the order of the mean cavity residence time. This finding limits the approach's applications in real-world scenarios. For example, the El Niño-Southern Oscillation (ENSO), which significantly influences regions like the Amundsen Sea (Paolo et al., 2018; Huguenin et al., 2024), may be poorly represented under accelerated forcing due to its typical 2-7 year cycle coinciding with the cavity residence time of certain ice shelves around Antarctica. Notably, cold-water shelves like the Fichner-Ronne Ice Shelf and the Ross Ice Shelf have a cavity residence time of 4-8 years (Nicholls and Østerhus, 2004; Loose et al., 2009), and warm-water shelves like those in the Amundsen Sea have even shorter cavity residence times given their smaller sizes and faster melting-driven cavity circulations. This alignment could lead to an overestimation of the melting response when ENSO's timescale is compressed under accelerated forcing. Moreover, even if the multi-decadal variation in forcing substantially exceeds the cavity residence time, applying the accelerated forcing approach may result in an underestimation of basal melting response once its compressed timescale is comparable to the cavity residence time. Therefore, caution should be used when applying the accelerated forcing approach to studies addressing climate variability on sub-decadal to decadal timescales.*"

We have also expanded the discussion section in the revised manuscript to address the reviewer's concern about the mixed timescale scenarios, as

*" For scenarios involving mixed timescales, such as seasonal forcing superimposed on decadal oscillations with a steady background increase, additional experiments are necessary to yield definitive answers. Addressing these complex interactions requires a broader range of studies to fully understand the dynamics at play. These studies would help clarify how various overlapping timescales influence each other, which is essential for more accurate climate modeling. Such investigations, however, fall beyond the scope of our current study and represent important directions for future research."*

*The authors provide a publicly accessible archive (similar as in Zhao et al. 2022, https://doi.org/10.5194/gmd-15-5421-2022) with detailed information about where to obtain the source code of ROMS, ElmerIce and the FISOC coupler (URLs + git commits) including configuration and restart files. As I've never worked with the described models, it is beyond my expertise to judge whether the given information and files are sufficient to reproduce the realized experiments. As far as I can tell the archive does not include information and restart files for the FVCOM model. I request the authors to check this, and update if necessary. Furthermore it would be helpful to include concrete information about the model versions and where to obtain the source code already in the manuscript, e.g. in the code availability section.*
*Also, please make sure, DOIs are provided for all references.*

We thank the reviewer point out this issue. We are now updated the code availability section, as

*"Code availability. The coupled model used the ice sheet model Elmer/Ice Version 9.0 (https://github.com/ElmerCSC/elmerfem. git;Gagliardini et al. (2013)), the ocean model FVCOM (https://dx.doi.org/10.17632/m6g4c3hm9m.1], Zhou and Hattermann (2020) ), the ocean model ROMSIceShelf Version:1.0 with code (https://doi.org/10.5281/zenodo.3526801; Galton-Fenzi (2009) ), and the coupled framework FISOC Version 1.1 (https://doi.org/10.5281/zenodo.4507182; Gladstone et al. (2021)). The FISOC-ROMSIceShelf-Elmer/Ice source code and input files needed to run the ROMS-based coupled experiments in this study are all publicly available (https://doi.org/10.5281/zenodo.5908713, Zhao et al. (2022)). The FISOC-FVCOM-Elmer/Ice model shares the same ice sheet model input files as ROMS. The FVCOM-based simulations use the same ice sheet model input files as the ROMS-based simulations. The input and output files for these FVCOM-based simulations are preserved at the Norwegian national research data archive and can be downloaded anonymously by anyone via a web-based interface. A DOI will be assigned upon acceptance of the manuscript."*

*Furthermore, I share the concern by Nicolas Jourdain (RC1) about how to deal with calving fluxes in more realistic/non-local applications.*

We have addressed the concern raised by Nicolas Jourdain regarding calving

fluxes and glacial meltwater input in more realistic applications. This discussion is now included in the revised manuscript, as follows:

[revised manuscript text omitted]
 146, 101536. URL: https://www.sciencedirect.com/science/article/pii/S1463500319301738, doi:https://doi.org/10.1016/j.ocemod.2019.101536.

---

## Author Response (AR2)

**Reviewer #1**

**General comments**

*The authors have done a great job at addressing most of my comments. There are nonetheless two points that still need to be improved:*

First of all, we would like to express our sincere gratitude to the reviewer for their thorough comments and constructive suggestions, which have greatly contributed to improving the quality of our manuscript. We also appreciate the reviewer's recognition of our efforts during the revision process, and we have learned a great deal through this experience. Next, we will address the two remaining points raised by the reviewer.

*1- To keep a realistic freshwater budget for the ocean in the accelerated approach, the authors suggest "applying periodic restoration techniques to adjust the ocean's salinity and temperature fields using observed or targeted values". For multi-centennial projections, which are identified as the ideal target for the accelerated approach, such restoring nonetheless requires the prior knowledge of temperature and salinity projections. Hence, the accelerated approach would only be applicable for a kind of downscaling of the CMIP simulations with an ice sheet-ocean model, not for a fully coupled climate model with interactive ice sheets. This should be mentioned in the discussion.*

We thank the reviewer for pointing out the limitation of 'applying periodic restoration techniques to adjust the ocean's salinity and temperature fields using observed or targeted values' in fully coupled climate models with interactive ice sheets. In lines 571-574 of the revised version of the manuscript, we have incorporated a discussion of this limitation, which states: "For multi-centennial projections, the ideal target for the accelerated approach, such restoration requires prior knowledge of temperature and salinity projections. As a result, the accelerated approach is most applicable for downscaling simulations from the Coupled Model Intercomparison Project (CMIP) using an ice sheet-ocean model, rather than for fully coupled climate models with interactive ice sheets."

*2- It is a problem that the abstract does not clearly state the caveats (challenges) of this approach. Currently, the abstract ends with "When appropriately applied, the accelerated approach can be a useful tool in coupled ice sheet-ocean modelling", which is not really demonstrated given the remaining questions on the mixed time scales (seasonal to climate trends) in realistic simulations and the associated challenge to close the ocean freshwater budget (see previous point).*

We apologize for the omission in the abstract. To address the caveats of the accelerated forcing approach in the abstract, we have replaced the sentence "When appropriately applied, the accelerated approach can be a useful tool in coupled ice sheet-ocean modelling" with "We have also discussed the limitations

of applying the accelerated forcing approach to real-world scenarios, as it may not be applicable in coupled modeling studies addressing climate variability on sub-decadal, decadal, and mixed timescales, or in fully coupled climate models with interactive ice sheets. Nevertheless, when appropriately applied, the accelerated approach can be a useful tool in process-oriented coupled ice sheet-ocean modeling or for downscaling climate simulations with a coupled ice sheet-ocean model." This revision has been made in the abstract of the revised version of the manuscript (lines 18-23).

**Reviewer #2**

**General comments**

*Dear editor Riccardo Farneti, dear author Qin Zhou and others,*

*I appreciate the changes made to the manuscript and congratulate the authors for the good paper. I feel that my comments and the ones of the other reviewers have been addressed sufficiently.*
*After responding to some comments, which are mostly of technical nature, I can recommend the manuscript for publication in GMD.*

*Best regards,*

*Moritz Kreuzer*

We are grateful for the reviewer's positive feedback on our revision and for the technical comments provided on the revised manuscript. We would also like to thank the reviewer for the valuable comments and suggestions from the previous round of review, which have greatly enhanced the quality of the manuscript. Below, we provide our point-by-point responses to the reviewer's technical comments.

**Specific Comments**

*-L. 83: "sensitive to the boundary conditions" - the authors wanted to change this to "sensitive to the timescale of varying boundary conditions"*

We have changed "sensitive to the boundary conditions" to "sensitive to the timescale of varying boundary conditions " in the revised version of manuscript.

*- l.143: "(ROMS, (Shchepetkin and McWilliams, 2005)" - one bracket too much*

Corrected.

*- Fig. 3: "FVCOM-ISMOP+" - change to "FVCOM-ISOMIP+".*

Corrected.

*- l.244: "blue lines" - I think this is supposed to be "cyan lines".*

Corrected.

*- l.255-259: - Percentages for cavity averaged melt rates given here seem to not exactly match with the values plotted in Fig. 6 (assuming the values are repre-*

*sented by the center of the blobs).*

Thank you for your observation. We apologize for any confusion caused. The numbers in the center of the colored circles in Figure 6 actually represent the period of the oscillating profiles used in each simulation, not the percentages for cavity-averaged melt rates. To clarify this and avoid misunderstandings, we have replaced " The period of oscillating profiles used in each simulation is denoted by the black text within the colored circles" with " Numbers in the colored circles denote the period of oscillating profiles used in each simulation." in the caption of Figure 6 in the revised version of the manuscript.

*- l.317: "1200 months" - I suggest to write "100 years" here, similar to the other time spans, that are given in years now.*

We have replaced "1200 months" with "100 years" in the revised version of the manuscript.

*- l.338 "over" - remove double occurrence.*

Removed.

*- l. 370 and 430: "Figure ??" - the text reference seems to be corrupted.*

Corrected.

*-L. 83: - Fig. 11a - Why is the melt rate around 1.4 m/yr? L. 323 states that Pfast experiments are restarted from FI_C2M simulation, which has a mean melt rate close to 2 m/yr (Fig. 4).*

The reviewer's observation is valid. The difference in melt rates between the FI_C2M and Pfast experiments is due to the use of different vertical Prandtl numbers (VPRNU) in the stand-alone and coupled model setups. In FVCOM, the vertical Prandtl number (VPRNU) is defined as the ratio of vertical thermal diffusion to vertical eddy viscosity. It is included in the thermal diffusion term in the temperature or salinity equation. A Prandtl number of 1 implies that turbulent mixing transfers heat and momentum equally, while a value less than 1 indicates reduced thermal diffusion compared to eddy viscosity.
In the stand-alone simulations, VPRNU was set to 1, resulting in a higher melting rate of close to 2 m/yr. In contrast, in the coupled simulations, VPRNU was set to 0.01, which reduces thermal diffusion and leads to the mean melt rate dropping to 1.4 m/yr a few days after initialization. Although these settings differ, they do not affect our study's conclusions, as we did not directly compare the coupled with the standalone simulations. Our conclusions are drawn from analyses within each experiment class independently, where the vertical Prandtl numbers were consistent within each class, ensuring that differences in Prandtl number do not impact the validity of our results.

*- l.443 and 445: "Figure 14" - Figure 13?*

Corrected.

*- l.510: "deviates slightly" - maybe add "only"?*

Added.

*- l.547: "ice(Bintanja" - add space.*

Added.

*- l.548: "Moorman et al., 2020), (Bronselaer et al., 2018;" - replace "), (" by
"; ".*

Replaced.

*- "Code availibility" - the embedded link to elmerfem github repository accidently
contains a space: "elmerfem .git".*

Corrected.

**Reviewer #3**

**General comments**

*I am satisfied with the authors response to my comments and appreciate the effort gone to in adding additional figures of grounding line movement. Provided the other reviewers are similarly satisfied I would be happy to recommend publication.*

We are pleased to know that the reviewer is satisfied with the revisions. We would like to express our gratitude once again for their insightful comments on the grounding line movement in the previous review round, which have significantly enriched our manuscript.